# Towards parametrising atmospheric concentrations of ice nucleating particles active at moderate supercooling

Claudia Mignani[a], Jörg Wieder[b], Michael A. Sprenger[b], Zamin A. Kanji[b], Jan Henneberger[b], Christine Alewell[a], and Franz Conen[a]

[a]Department of Environmental Sciences, University of Basel, Bernoullistrasse 30, 4056 Basel, Switzerland
[b]Institute for Atmospheric and Climate Science, ETH Zurich, Universitätstrasse 16, 8092 Zurich, Switzerland

**Correspondence:** Claudia Mignani (claudia.mignani@unibas.ch) or Franz Conen (franz.conen@unibas.ch)

**Abstract.** A small fraction of freezing cloud droplets probably initiates much of the precipitation above continents. Only a minute fraction of aerosol particles, so-called ice nucleating particles (INPs), can trigger initial ice formation at $-15$ °C, at which cloud-top temperatures are frequently associated with snowfall. At a mountain top site in the Swiss Alps, we found that concentrations of INPs active at $-15$ °C can be parametrised by different functions of coarse (> 2 µm) aerosol particle

concentrations, depending on whether an air mass is (a) precipitating, (b) non-precipitating, or (c) carrying a substantial fraction of dust particles while non-precipitating. Consequently, we suggest that a parametrisation at moderate supercooling should consider coarse particles in combination with air mass differentiation.

## 1 Introduction

The presence of ice in clouds is important for precipitation initiation (Mülmenstädt et al., 2015; Heymsfield et al., 2020). Ice nucleating particles (INPs) affect clouds and their development by generating primary ice at temperatures between 0 and $-38$ °C. The difficulty of understanding and thus predicting atmospheric INP concentrations ([INP]) originates from observational challenges related to field measurement techniques (Cziczo et al., 2017), the large variety of potential sources (Kanji et al., 2017), and the wide range in atmospheric abundances from $10^{-6}$ to $10^3$ L$^{-1}$ (Petters and Wright, 2015). The past decade

has seen substantial efforts toward improving empirical parametrisations of [INP] (e.g., DeMott et al., 2010; Phillips et al., 2013; DeMott et al., 2015). A current empirical parametrisation established by DeMott et al. (2015), hereafter D15, predicts [INP] based on nucleation temperature and number concentration of mineral dust particles with diameters > 0.5 µm ($[n_{0.5}]$) (DeMott et al., 2015). Although the D15 may be applicable to temperatures below $-20$ °C, it is not expected to represent a multivariate INP population and remains "weakly constrained at temperatures $>-20$ °C, where much additional ambient and

laboratory data are needed" (DeMott et al., 2015).

The coldest part of a cloud - typically cloud-tops and their temperature - determines what fraction of the INP population will get activated and form ice crystals. Any INPs with colder activation temperatures will remain inactive. Cloud-top temper-

atures associated with winter snowfall have a primary temperature mode near $-15\,°C$, as derived from close to $10^5$ parallel observations of cloud-top temperatures and falling solid precipitation throughout the United States (Hanna et al., 2008). The majority of these observations were for light snowfall. In contrast, cloud-top temperature distributions for moderate and heavy snowfall are bimodal with a second, minor mode around $-40\,°C$ (Hanna et al., 2008). This is consistent with observations in mountainous regions (Rauber, 1987). Of all snowfall observations with cloud-tops above homogeneous freezing temperature (i.e. $>-38\,°C$), approximately 30% are associated with cloud-top temperatures not colder than $-15\,°C$ (Hanna et al., 2008). Therefore, substantial fractions of initial ice crystals in snow-producing mixed-phase clouds may be caused by INPs that nucleate ice at temperatures $\geq-15\,°C$ ($INP_{s-15}$). This inference may extend to other midlatitude continental regions. Considering $-15\,°C$ as a temperature that is important for snow formation also makes physical sense, because maximum depositional growth of ice crystals is around $-15\,°C$ (Rogers and Yau, 1989). We, therefore, in this work, focus on INPs active at that temperature, although future studies would benefit from relating measurements to overall cloud thermal structures, which may at times include lower cloud top temperatures.

Based on current understanding, atmospheric $INP_{s-15}$ are mostly biological aerosol particles (Murray et al., 2012). Although their number concentration is generally smaller compared to those $<-15\,°C$ (Petters and Wright, 2015), primary ice formed by $INP_{s-15}$ may get multiplied by an order of magnitude due to secondary ice formation (Mignani et al., 2019). Findings from a sparse number of size-resolved measurements of atmospheric INPs show that $INP_{s-15}$ are mostly $>2\,µm$ in diameter (Huffman et al., 2013; Mason et al., 2016; Creamean et al., 2018). This particle size is however under-represented for instrumental reasons in the empirical data on which D15 and other parametrisations (e.g., DeMott et al., 2010; Phillips et al., 2013) are based. Furthermore, an increase in atmospheric abundance of INPs active at moderate supercooling have been observed during precipitation (Bigg and Miles, 1964; Huffman et al., 2013; Hara et al., 2016; Conen et al., 2017). This might be explained by aerosolisation of INPs by rain itself, a mechanism similar to the generation of bioaerosol by raindrop impingement (Joung et al., 2017), which is probably dependent on various parameters like surface wetness or land cover.

To test whether the general approach of D15 (i.e. parametrising INPs as a function of particles larger than a certain size) can be reconciled with the findings of $INP_{s-15}$ being mostly larger than $>2.0\,µm$ and increasing during precipitation, we collected and analysed aerosol samples from February to March 2019 on Weissfluhjoch, Switzerland at average local air temperature of $-7.1$ (s.d.$\pm4.3$) $°C$ during sampling intervals (Fig. S1). The site, surrounding mountains and nearby valleys were snow-covered, while most of the lower lying plain and the foothill regions were not, and precipitation occurred in the form of rain in those regions during our study period.

## 2  Material and Method

Between 11 February and 26 March 2019, we collected and analysed a total of 140 aerosol samples at Weissfluhjoch, Switzerland (46°49'58.670" N, 9°48'23.309" E, 2671 m a.s.l.) during the "Role of Aerosols and Clouds Enhanced by Topography on Snow (RACLETS)" campaign. Total aerosol was sampled through a heated inlet (heating element kept at $+46\,°C$) similar to the one described in Weingartner et al. (1999), which was designed such that particles with diameters $<40\,µm$ are sampled

up to a wind speed of 20 m s$^{-1}$. The inlet extended through the eastern wall of the laboratory and was about 8 m above local ground. The aluminium inlet tubing had an inner diameter of 4.5 cm throughout its total length of 7 m. Particles entering the inlet travelled at a speed of about 3 m s$^{-1}$ first 2.5 m downward, then turned by 70° in a radius of 20 cm towards the inside of the laboratory and continued for another 4.5 m about 20° downslope before being trapped in the impinger, approximately 2.2 s after they had entered the inlet. Ice particles resuspended from surrounding surfaces (snow-covered throughout the campaign and with average local wind speed of 7.1 (s.d.±3.4) m s$^{-1}$ during sampling intervals) cannot be ruled out, but are unlikely to contribute significant amounts to the total sampled particles. The air flow was maintained throughout the campaign at 300 L min$^{-1}$, during sampling by a high flow-rate impinger (Bertin Technologies, Coriolis®µ) and between sampling intervals by a makeup flow using an external blower. In addition, an Aerodynamic Particle Sizer (APS; Model 3321, TSI Corporation) sampled from the same inlet upstream of the impinger at 1 L min$^{-1}$.

Aerosol samples were collected using the Coriolis®µ as was done in recent studies (Els et al., 2019; Tarn et al., 2020; Miller et al., in review, 2020). Each sample consisted of aerosol particles collected throughout 20 min (i.e. from 6 m$^3$ of air) into 15 mL of ultra-pure water (Sigma-Aldrich, W4502-1L). With increasing particle size the theoretical sampling efficiency of the Coriolis®µ increases from around 50% for particles of 0.5 µm in size, 80% for particles of 2.0 µm, to close to 100% for particles of 10 µm (personal communication with Bertin Technologies). Water losses due to evaporation were compensated by replenishing the circulating water after 10 and 20 min. To avoid eventual storage effects (Beall et al., in review, 2020), samples were analysed immediately after collection in a drop freezing assay, with 52 droplets of 100 µL each, as previously described (Stopelli et al., 2014) and cumulative [INP] were calculated (Vali, 1971). Sampling and analysis were designed in such a way that expected [INP$_{-15}$] of each sample would be well within the detection limits, meaning that several but not all droplets in the assay would be frozen. With our sampling and analysis design the detection range lies between 4.8x10$^{-4}$ (i.e. first drop frozen) and 8.1x10$^{-2}$ L$^{-1}$ (i.e. second last drop frozen). In 15 samples, all dropets were frozen and in one sample no droplet was frozen at −15 °C. These samples were not considered because their [INP$_{-15}$] were outside the detection limits. For the other samples (n = 124) several, but not all droplets froze. Background measurements (n = 15) following identical procedure as with the samples, but without turning on airflow of the impinger, were below detection limit. Number concentrations of particles [n] with aerodynamic diameters from 0.5 µm to 20 µm were measured with the APS (20 s scanning time), and were integrated (summed) from the particles sizes of interest onward i.e. ≥ 0.542 µm for [n$_{0.5}$] (51 bins) and ≥ 1.982 µm for [n$_{2.0}$] (33 bins). The 20 s data was averaged over each time-period (20 min) of the taken impinger-based aerosol samples. [n$_{0.5}$], [n$_{2.0}$] and [INP$_{-15}$] were adjusted to standard pressure conditions (std; p$_{ref}$ = 1013.25 hPa). [INP$_{-15}$] estimates based on D15 were calculated as:

$$INP_T = cf \cdot n_{0.5}{}^\beta \cdot e^{\gamma \cdot (-T) + \delta} \tag{1}$$

where $\beta = 1.25$ , $\gamma = 0.46$, $\delta = -11.6$, $T$ is the temperature in degree Celsius, $INP_T$ the ice nucleation particle concentration (std L$^{-1}$) at $T$, and $n_{0.5}$ the number concentration of aerosol predominantly consisting of mineral dust particles with a physical diameter > 0.5 µm (std cm$^{-3}$). A physical diameter of 0.5 µm is equivalent to an aerodynamic diameter of 0.9 µm,

assuming a particle density of 2.6 g cm$^{-2}$ and a shape factor of 1.3 (Raabe, 1976), which are typical values for mineral dust particles. Similar transformations for observations not dominated by mineral dust would require information about densities and shapes of the main components of sampled particle populations, which were not available for our site and would require unsupported assumptions. Therefore, we chose to show for all our observations the directly measured particle concentrations in terms of *aerodynamic diameter*. To use D15 parametrisation in our context, we corrected predicted [INP] for the difference between the aerodynamic diameter measured and the physical diameter used in Eq. 1 by multiplying $n_{0.5}$ in Eq. 1 by the ratio of particles with aerodynamic diameter > 0.9 μm (equivalent to 0.5 μm physical diameter) to particles with aerodynamic diameters > 0.5 μm, which we observed in Saharan dust dominated air masses during our campaign. The average value of this ratio was 0.59. The calibration factor $cf$ accounts for so-called instrument-specific calibration and is suggested to be three ($cf$ = 3) to predict maximum immersion mode atmospheric [INP] (DeMott et al., 2015). Schrod et al. (2017), who collected samples with an unmanned aircraft system in the Mediterranean region with substantial Saharan Desert dust influence, used it as a mathematical degree of freedom when fitting Eq. 1 to their observations.

Five-day back trajectories were calculated using the Lagrangian analysis tool LAGRANTO (Sprenger and Wernli, 2015). For each sample, one trajectory was started at the full hour closest to the sampling time and from the exact sampling position. The driving wind fields were taken from the operational analysis of the Swiss National Weather Service (COSMO1; www.meteoswiss.ch) and the European Centre for Mid-Range Weather Forecasts (ECMWF; www.ecmwf.int). Started in the COSMO domain, the trajectories were extended based on ECMWF data at the time and location where they leave this domain. Their position was saved every 10 min. Along the trajectories, total precipitation was traced amongst others (i.e. height, pressure, temperature, specific humidity and surface height) enabling us to determine the total precipitation amount along the last 6 hours prior to sampling (Fig. S2).

## 3 Results and Discussion

We found cumulative concentrations of atmospheric INPs active at −15 °C ([INP$_{-15}$]) that are lying within the lower half of values summarised in Petters and Wright (2015). From the total of 124 impinger-based aerosol samples with quantified [INP$_{-15}$], about half (56) were collected from air masses that had precipitated at least 1.0 mm during the 6 hours prior to sampling (defined as "precipitating"). About half of these air masses were also precipitating when sampled at Weissfluhjoch, as observed by a precipitation gauge. A similar number of samples (57) were from air masses with less or without any prior precipitation ("non-precipitating") and 11 were from air masses including a substantial fraction of Saharan dust (SD) and no prior precipitation. Air masses were mainly reaching the sampling position from the West (Fig. 1a). Precipitating air masses were coming on a rather direct way from the Atlantic crossing the West of Europe with less detour than non-precipitating air masses whereas air masses carrying dust where coming from the direction of the Saharan desert passing the South of Europe. Six hours before arriving at Weissfluhjoch, the trajectories crossed a mean distance of 242 (s.d.±145) km and spent two third of the time over Switzerland (Fig. 1b). Forested land (31%), agricultural fields (17%), pasture (12%) and natural grasslands

(12%) were the most common land covers they were passing, as derived by the European Copernicus programs Corine land cover map (https://land.copernicus.eu/pan-european/corine-land-cover/clc2018 (access: 13 October 2020)).

Precipitating air masses had the lowest $[n_{0.5}]$ and the lowest concentration of aerosol particles with diameters > 2 μm ($[n_{2.0}]$), but similar ratios as non-precipitating air masses of $[n_{2.0}]$ to $[n_{0.5}]$ (Fig. 2a-b). The largest ratio of $[n_{2.0}]$ to $[n_{0.5}]$ was in SD air masses (Fig. 2c). Therefore, relative differences in measured $[INP_{-15}]$ between precipitating and non-precipitating air masses would be affected very little, if a substantial fraction of INPs$_{-15}$ would have been of a size near 0.5 μm, which was sampled with a lower efficiency (50%) than 2 μm (80%). However, $[INP_{-15}]$ in both of these air masses would have been underestimated relative to $[INP_{-15}]$ in SD affected air masses, which had the highest $[n_{2.0}]$ to $[n_{0.5}]$ ratio.

In general, $[INPs_{-15}]$ in non-precipitating and precipitating (not dominated mineral dust) air masses were higher than in mineral dust dominated air masses for the same $[n_{0.5}]$ (Fig. 3a). The observed slope for SD air masses was the same as that predicted by the D15 parametrisation. The offset of the D15 curve depends on the calibration factor ($cf$, Eq. 1). Observed SD data were between the D15 curves with $cf$ set to 1 and to 0.086, respectively. The latter value is reported in Schrod et al. (2017), who sampled the Saharan Dust Layer above Cyprus with a drone up to 2850 m a.s.l.

Considering the fact that the observed size of INPs$_{-15}$ is mostly larger than 2 μm (Huffman et al., 2013; Mason et al., 2016; Creamean et al., 2018), we plot measured $[INPs_{-15}]$ against $[n_{2.0}]$, instead of $[n_{0.5}]$, resulting in a more distinct separation of the data to the different air masses (Fig. 3b). In each of the three categories of air masses, $[INP_{-15}]$ can be described as a function of $[n_{2.0}]$ that is valid for a range of $[INP_{-15}]$ from 0.0006 std L$^{-1}$ to 0.14 std L$^{-1}$ (Table 1). For data in SD and non-precipitating air masses, $[INP_{-15}]$ can be described as power functions of $[n_{2.0}]$ with similar linear slopes on a log-log plot, but lower $[INP_{-15}]$ per $[n_{2.0}]$ for SD. This goes hand in hand with the earlier observations that air masses influenced by SD carry less INPs active at moderate supercooling per unit mass of aerosol particles than European background air masses (Conen et al., 2015). In precipitating air masses, the ratio between $[INP_{-15}]$ and $[n_{2.0}]$ was usually larger than in non-precipitating air masses. This reveals that the aerosol population was enriched with INPs active at moderate supercooling during precipitation, consistent with previous findings (Bigg and Miles, 1964; Huffman et al., 2013). Since additional INPs during precipitation might be due to aerosolisation of INPs by rain which is likely independent of the background in [n], we describe $[INP_{-15}]$ in precipitating air masses by adding a constant to the function fitted to the non-precipitating cases (Fig. 3b). The median difference between the function of non-precipitating air masses and measured $[INP_{-15}]$ in precipitating air masses was 0.014 std L$^{-1}$ (Fig. 4). The relationship between these differences and $[n_{2.0}]$ was weakly positive and not significant, meaning that the absolute value of additional INPs in precipitating air masses was independent of $[n_{2.0}]$. This finding corroborates our assumption that additional INPs during precipitating air masses are independent of background [n]. A consequence of our finding is that to precipitating air masses with low $[n_{2.0}]$, the addition of INPs aerosolised by precipitation makes a relatively large contribution to the overall $[INP_{-15}]$. The additional INPs during precipitation could be emitted through the impact of rain on snow-free lower lying plain regions, a speculation which needs to be investigated in future.

Overall, for each air mass class, the correlation coefficient of the obtained functions is equal or higher with $[n_{2.0}]$ as a predictor than with $[n_{0.5}]$ (Table 1). This confirms that $[n_{2.0}]$ is a more powerful predictor of INPs$_{-15}$ than $[n_{0.5}]$ when combined with air mass differentiation (Fig. S3). It underlines the importance to consider aerosol particles > 2 μm. To further develop a

parametrisation valid for temperatures $> -20$ °C, we suggest to further investigate the presented functions, because INPs active at other temperatures or at other locations and during different seasons may also be associated with other particle sizes or other INP concentrations. Especially the addition of INPs in precipitating air masses should be constrained with data from all over the globe.

## 160 4  Conclusions

In summary, it is possible to reconcile two fundamental aspects of INPs active at moderate supercooling - increased abundance during precipitation and size - with a widely used approach to parametrise INPs active at colder temperatures. Parametrisations based on the number concentration of aerosol particles are reasonable to predict INPs at moderate supercooling. However, relating [INPs$_{-15}$] to the number concentration of larger particles can further improve the predictions, which is not to say that INPs$_{-15}$ are always in such size range. An even greater improvement in predictions is possible when we additionally distinguish between air masses that are precipitating, non-precipitating, or carrying a substantial fraction of Saharan dust. More of the variance of INP concentrations was explained by aerosol concentrations in air masses that were non-precipitating or carrying desert dust as compared to air masses that were precipitating. The absolute value of additional INPs in precipitating air masses, versus non-precipitating air masses, seems to be independent of total aerosol concentrations.

To tackle predictions of INPs active at moderate supercooling, particular attention has to be attributed to sample larger aerosol particles at mixed-phase cloud height, including air masses that have been precipitating, and adjust procedures to reliably quantify [INP] at the targeted activation temperatures, as was done in this study. Although our proposed parametrisation has a generalisable structure, the parameters are so far only constrained by data from one campaign. While a new parametrisation for a previously weakly constrained temperature is clearly beneficial, it complements rather than replaces previous parametrisations. In a changing climate, with increasing temperatures and changing precipitation patterns, it is important to predict feedbacks between INPs and precipitation.

## 5  Data availability

https://www.envidat.ch/dataset/ice-nucleating-particle-concentrations-active-at-15-c-at-weissfluhjoch

*Author contributions.* CM and FC conceived the study. CM and JW conduced the measurements. JW provided the aerosol data. MAS conduced the modeling. JH and ZAK hosted the entire measurement campaign. CM processed the data and prepared the figures of the manuscript. CM and FC interpreted the data and wrote the manuscript with contributions from JW, MAS, ZAK, JH and CA.

*Competing interests.* The authors declare no competing interest.

*Acknowledgements.* The authors would like to deeply thank Paul DeMott and the anonymous Referee for their careful reviews. We are indebted to Martin Genter for logistical support, Carolin Rösch and Michael Rösch for technical support, Nora Els for borrowing the Coriolis®µ of University of Innsbruck as well as sharing her experience and Lucie Roth and Mario Schär for helping with the measurements. We thank MeteoSwiss for weather data and providing access to the COSMO1 and ECMWF data and all the RACLETS members for fruitful discussions. JH, JW and ZAK as well as CM and FC acknowledge funding from the Swiss National Science Foundation (SNSF) grant numbers 200021_175824 and 200021_169620, respectively.

185

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

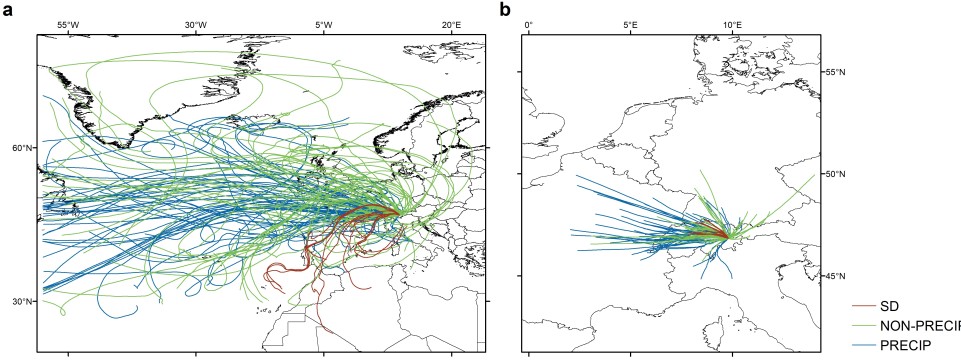

**Figure 1.** (**a**) Five-days and (**b**) six-hours back trajectories of air masses that were precipitating (PRECIP, blue), non-precipitating (NON-PRECIP, green), and carrying a substantial fraction of Saharan dust while non-precipitating (SD, red).

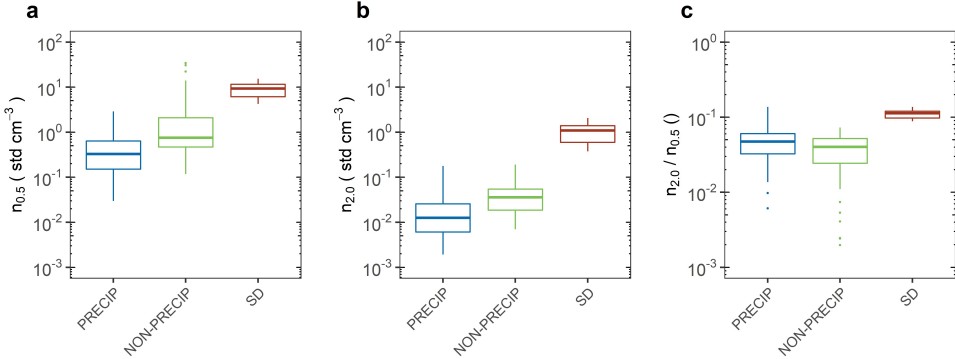

**Figure 2.** Number concentrations of aerosol particles with aerodynamic diameters (**a**) > 0.5 µm [$n_{0.5}$] and (**b**) > 2.0 µm [$n_{2.0}$] and (**c**) their ratio for the aerosol populations of PRECIP, NON-PRECIP and SD air masses.

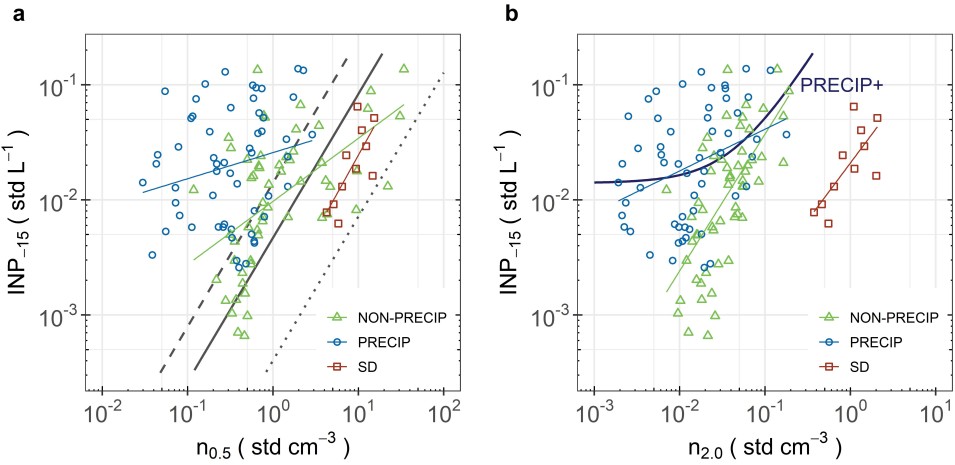

**Figure 3.** Cumulative concentrations of ice nucleating particles active at −15 °C [INP$_{-15}$] (**a**) versus [n$_{0.5}$] and (**b**) versus [n$_{2.0}$] for PRECIP (blue circles), NON-PRECIP (green triangles), and SD (red squares) air masses. Power functions (solid lines) for each type of air masses based on [n$_{0.5}$] and [n$_{2.0}$] are shown. An additional preliminary parametrisation for precipitating air masses based on [n$_{2.0}$] is shown (PRE-CIP+, thick dark blue line). It is the same as for non-precipitating air masses but with an added constant equivalent to 0.014 INPs L$^{-1}$. The corresponding equations and R$^2$ values are shown in Table 1. The gray lines show the D15 parametrisation extrapolated to −15 °C and corrected for the difference between physical and aerodynamic diameters (see Method section) with three different calibration factors (Eq. 1): $cf$ = 3 (dashed), $cf$ = 1 (continuous), and $cf$ = 0.086 (dotted). The latter value was the best fit found by Schrod et al. (2017) for Saharan dust above Cyprus.

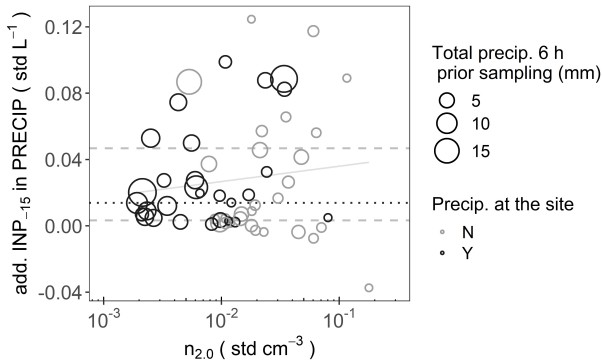

**Figure 4.** Difference between [INP$_{-15}$] in precipitating (PRECIP) air masses and the function fitted to the non-precipitating (NON-PRECIP) air masses (additional [INP$_{-15}$] in PRECIP) versus [n$_{2.0}$]. The median difference is 0.014 std L$^{-1}$ (black dotted line) and the lower and upper quartiles are 0.003 and 0.047 std L$^{-1}$, respectively (grey dashed lines). The linear fit (grey solid line) is weakly positive but not significant (Pearson correlation test, R = 0.0027 and p = 0.98). Circle area is proportional to the amount of precipitation along the last 6 hours of the trajectory prior to sampling. Black circles are for samples precipitating at Weissfluhjoch.

**Table 1.** Equations of the functions shown in Fig. 3 (i.e. PRECIP, PRECIP+, NON-PRECIP, SD) predicting cumulative concentrations of ice nucleating particles active at $-15\ ^\circ$C [INP$_{-15}$] based on aerosol particles with aerodynamic diameters $> 0.5\ \mu$m [n$_{0.5}$] and $> 2.0\ \mu$m [n$_{2.0}$] and their respective $R^2$ values. In addition, equations and $R^2$ values of power functions fitted to all data points irrespective of air mass classes are shown (ALL). The equations are listed based on the following formula: $y = b \cdot x^a + c$, with y equal to [INP$_{-15}$].

| Air mass type | n | $x$ | $b$ | $a$ | $c$ | $R^2$ |
|---|---|---|---|---|---|---|
| ALL | 124 | [n$_{0.5}$] | 0.02 | 0.19 | 0 | 0.06 |
| PRECIP | 56 | [n$_{0.5}$] | 0.03 | 0.23 | 0 | 0.05 |
| NON-PRECIP | 57 | [n$_{0.5}$] | 0.01 | 0.55 | 0 | 0.29 |
| SD | 11 | [n$_{0.5}$] | 0.001 | 1.34 | 0 | 0.55 |
| | | | | | | |
| ALL | 124 | [n$_{2.0}$] | 0.03 | 0.22 | 0 | 0.07 |
| PRECIP | 56 | [n$_{2.0}$] | 0.09 | 0.36 | 0 | 0.12 |
| PRECIP+ | 56 | [n$_{2.0}$] | 0.58 | 1.19 | 0.014 | 0.14 |
| NON-PRECIP | 57 | [n$_{2.0}$] | 0.58 | 1.19 | 0 | 0.44 |
| SD | 11 | [n$_{2.0}$] | 0.02 | 0.99 | 0 | 0.55 |