# Peer review of "Towards parametrising atmospheric concentrations of ice nucleating particles active at moderate supercooling"

_Atmospheric Chemistry and Physics, 2020_

## Referee Comment (RC1) · Paul DeMott (Referee) · 28 Jul 2020

**General Comments**

This is a welcome and concise contribution to the literature regarding parameterization of ice nucleating particle measurements. It suggests that the size required for best relating INP concentrations to aerosols deserves revisiting in general, depending on the conditions for INP processing, and it suggests that a larger size is required than used in past parameterizations using aerosol concentration-size relations as a parameterization basis (at least for the mineral dust related parameterization they investigate). This is not an entirely new recommendation, but is clearly presented here. I do not subscribe

to $-15°C$ as the critical or only temperature needed for constraining parameterizations as relevant to wintertime precipitation. In general, the non-comprehensiveness of using only $-15°C$ data is a limitation for any widespread application. I have some comments as well on clarifying the specification of size as being aerodynamic versus physical. This matters for the comparison to literature shown. In that regard, it should be made clear that the D15 parameterization is specifically for mineral dusts. The findings regarding the relation or not of apparent biological INPs to aerosol sizes is a new one to my knowledge, and is very interesting in its implications. This paper clearly motivates future studies to improve parameterizations overall through use of field data.

**Specific Comments**

1) Abstract: Lines 5-6: A key question for this might also be how divorced a surface site is from the free troposphere where clouds form? Or are the clouds always tied to the boundary layer and have cloud tops relevant to the measurements.

2) Introduction: Line 13: To be clear, this parameterization is for mineral dust as a single category only. It should not be expected to represent a multivariate INP population.

Lines 17-19: While this dive into the meteorology/climatology of winter precipitation is much appreciated, I want to note a caveat with regard to these studies. First, they were dominated by observations from over continental regions that were largely non-mountainous, unlike the site in the present study. Secondly, colder cloud top modes for precipitation are also noted in the referenced studies. This is consistent with earlier work by Rauber et al. (1987) from over mountainous regions in the United States, where both the $-15°C$ maximum associated with the dendritic growth regime and colder cloud top precipitation events are noted. There is no way to know when colder-topped clouds may be impacting the persistence of liquid water in the $-15°C$ regime, and thereby altering the microphysical scenario. Schultz et al. (2001) critiqued the inadequacy of this isotherm alone (in the Wetzel et al. scheme) for predicting heavy precipitation that can occur over a broad range of cloud top temperatures. So perhaps

note this as "one of the modes" for winter precipitation. Hence, while it is reasonable to select this temperature for the present study, a caveat is needed because it is not possible to sit in any one location to make measurements and assume that the only relevant temperature for clouds passing over is a single value.

Line 20: What does the statement "$-15°$C or warmer" mean exactly? It is repeated in the Fig. 1 caption, but the "or warmer" part is not explained or justified. Also, with regard to the statement on the sizes of INPs relevant at this temperature, I would note that widespread measurements over continents in winter are, however, limited. Measurements of size-resolved INPs are even rarer in winter, at least to my knowledge of locations and the number of measurements that have been made of INP sizes. I understand that larger sizes indicate a better correlation in this paper, but I suggest care in assuming that the role of only larger INPs in this temperature regime is confirmed as relevant to winter clouds.

3) Material and Method

Line 33: Has the Coriolis collection method been compared to any other standard method, such as filters? This would be good to know.

Line 36: Can you clarify why there is an upper bound to measured concentrations?

Line 37: Does the detection limit mean the stated lower limit of quantification or what does it mean? I assume the background comparison would use particles per ml, rather than per L of air. Can you state values in that manner? Does it literally mean that no wells froze when you removed ultrapure water from the sampler and ran the same cooling rates?

Line 39: I am curious about the use of specific APS channel bin limits to define [n0.5] and [n2.0]. Aerodynamic size is different from physical size, and if using APS data to constrain values from a parameterization, especially D15, it requires conversion first to equivalent spherical physical diameter (using effective density or density and an

assumed shape factor), and then recalculation of the concentrations above 500 nm diameter. Otherwise the concentrations could be biased high for use in D15.

Line 46: Can you clarify what is meant by "amongst others" used here?

4) Results and Discussion

Line 59: As mentioned already, please clarify how [n0.5] as defined by the APS is used in D15. Also, I infer from Fig. 1 that you have not used the scaling factor derived in D15 for application in modeling total INP concentrations, but you have not stated that. It means that you do not assume the correction of continuous flow diffusion chamber data that was demonstrated valid for laboratory and transported ambient Saharan dust in D15. Please say so and perhaps justify.

Note on Figure 1: The listing of correlation coefficients for the fits shown, and perhaps also for the 500 nm concentrations (could list all of these in a table, for example), would be useful. This would put some quantification into the statement about "clarity of separation" of data when 2 microns is used as the reference. Also, the diamonds in the figure are extremely hard to resolve, but perhaps at a full page size the figure will be easier to read. Finally, please clarify again in the caption if the x-axis concentration values are for aerodynamic or physical size.

Line 68-69: Although I infer that you are getting at the enrichment of INPs that are inferred to be of biological origin following rain events, the statement "enrichment of the aerosol population with highly efficient INPs during precipitation" was a little confusing, because your data show that aerosol concentrations under rain events are typically lower on average.

Line 70: suggest "masses" for "mass"

Lines 73-75: While you have used a constant here to describe an assumed physical process whereby the production is independent of aerosols already in the air, I am curious if there is any dependence of this process on precipitation rate? Or in a complex

way on that and existing surface wetness? Does it always produce large INPs?

Line 90: As stated earlier, I do not feel that such a strong statement can be made regarding the importance primarily of the $-15°C$ INPs. Not without specific inspection for many sites around the world. And I believe that literature supports that this single temperature is not exclusive for predicting wintertime precipitation.

**References**

Rauber, R. M., 1987: Characteristics of cloud ice and precipitation during winter storms of the mountains of Northern Colorado, J. Clim. Appl. Meteor., 26, 488-524.

Schultz, D. M., Cortinas, J. V. Jr., and Doswell C.A. III, 2001, Comments on "An operational ingredients-based methodology for forecasting midlatitude winter season precipitation", Wea. and Forecasting, 17, 160-167.

---

## Referee Comment (RC2) · Anonymous Referee #2 · 31 Jul 2020

Review of "Towards parameterizing atmospheric concentrations of ice nucleating particles active at moderate supercooling" by Mignani et al.

General comments: This reviewer enjoyed reading this concise paper. The study presents several important aspects, such as precipitation INPs & their source appointment & implication of supermicorn INPs, that all could be an important addition to ACP. While the manuscript is overall well written, some sections deserve more articulations in this reviewer's opinion. See specific/technical comments given below. The reviewer hopes these help the authors and improve the manuscript.

The reviewer sees some speculative sentences (e.g., P3L71-75, P3L80-82). Concern-

ing these, at the end, claims of novelty and priority (P3L85-P4L91) are questionable to this reviewer. The reviewer understands that the parameterization of cumulative INPs @ -15 dC for 1.5 months is a snapshot example of the authors' concept/proposal of IN parameterization for moderate spercooling. Regardless, an elaborated summary on how specifically beneficial the proposed parameterization would be to the community seem necessary in the conclusion section.

The authors may want to soften the tone regarding the comparison to D15 etc., and carefully re-formulate all the associated statements. The take-home message of the paper should not be 'a previous parameterization does not work' (this reviewer felt this to some extent). Instead, the authors should structure the discussion by blending/merging previous contributions in this work (rather than clipping) and emphasize how their new parametrization can complement the previously introduced one(s). Plus, the reviewer questions how meaningful INP measurements at single T (-15°C) would be concerning many speculations and absurd statements given in the manuscript. The reviewer is aware that the justification of this single T point is given in P1L15-17. Nonetheless, the authors need to soften their tones in some sections.

Specific and technical comments:

P1L2: –> . . .at -15 °C, at which a cloud-top. . .

P1L3-4: –> We found at a mountain top site in Swiss Alps that INPs active at -15 °C contained different fraction of coarse (>2 $\mu$m) aerosol particles, depending on . . .

P1L10-11: If the authors are discussing about INPs in general (not droplet freezing in particular), then the water saturation should be one of the most important factors/variables to complicate our understanding in [INP] and should be included in this particular sentence. Plus, there should be more comprehensive and appropriate references to cite and support this sentence other than what appears here.

P1L13-14: Although successful. . . –> Although the D15 parameterisation may be ap-

plicable to temperatures below -20 °C, it remains. . .

P1L18: RE "it makes physically sense for various reasons" – then, the reviewer suggests the authors to briefly, at the least, discuss physical mechanisms of why it peaks specifically at -15 °C for the readers. Otherwise, -15 sounds like a magic number, making the manuscript and its concept sound speculative, in the reviewer's opinion.

P1L19-21: –> Recent findings from. . .; Creamean et al., 2015). However, a particle size is poorly represented in the empirical data-based parameterizations (Phillips et al., 2013). Perhaps, this part reads better this way?

P1L22: –> such high temperature active INPs

Section 2: How turbulent was it while sampling during snow precipitation? Some discussion of local meteorology and its impact on sampling/measurements beyond crude categorization based on observation seem necessary.

P2L26-27: –> While the site, surrounding mountains and nearby valleys were snow-covered, most of the foothill regions were not, and only rain persisted in those regions during our study period. Is this what the authors meant?

P2L31: The reviewer requests the authors to describe their inlet configuration in the manuscript. Was it a TSP isokinetic inlet? What was the height of the top of the inlet? Particle loss/transmission well characterized? How did the authors make sure there is no re-suspension of snow or soil got into the inlet as well as no influence of local gust/turbulence and other dynamic/thermodynamic effects?

P2L32: The reviewer requests the authors to elaborate their impinger particle caption efficiency in the manuscript. To the reviewer's knowledge, an impinger is good at capturing relatively large size particles, but not that efficient on trapping small particles. How did this kind of size-dependent trapping efficiency potentially affect the sampling activity and overall results should be addressed in the text.

P2L33: Perhaps some water were sucked up by a pump rather than being evaporated?

Then, replenishing pure water may have affected C_INP and n_INP estimations at the end? The authors presume only water evaporated and all aerosol particles remained in an impinger jar throughout individual samplings? Would it be really the case for the impinger, which was used in this study?

P2L36: Please clarify what "15 above, 1 below range" means.

P2L38: APS only appears once... No abbreviation seems needed.

P2L37-39: Number concentrations of particles should have been integrated rather than being averaged, correct?

P2L40: [n] and [INP] also scaled to standard T (273.15 K)???

P2L49: aerosol samples –> impinger samples

P3: General suggestion – the reviewer suggests the authors to discuss how their INP-15 generally compares to other, previous precipitation INP studies (e.g., Petters and Wright, 2015; https://doi.org/10.1002/2015GL065733) before jumping onto nX vs. INP-15. The reviewer understands that the authors intended to be straight on the point (and appreciate the concise, right on the point manuscript length to some extent), but the readers would appreciate this extra information to generalize/digest information at their end, in the reviewer's opinion.

P3: How SD-rich IN efficiency compares to Ullrich et al., 2017 or Niemand et al., 2012? The authors can estimate n_s and do comparisons?

P3L68-69: The reviewer is lost on the "It also reveals..." part. Please clarify what it means in an intuitive manner.

P3L71-75: Speculative sentences – Many questions came to the reviewer's mind - What was the influence of local thermodynamics & meteorology (esp. wind spd.)? Was a proper inlet used to eliminate the impact of local turbulence etc.? Chance of re-suspended particles getting into an impinger while high-volume sampling? Any hindsight 20/20 situations?

P4L87-88: Cannot disagree – a wider spatiotempral coverage is indeed needed.

P7: Concentrations of ice nucleating particles active at -15°C or warmer [INP-15] –> Cumulative INP concentrations estimated at -15°C, [INP-15]

Fig. 1: Show correlation coefficients for fits. Add fits & Rs in Fig.1g, too. Discuss these in P7.
* * *

---

## Author Comment (AC1) · 13 Oct 2020

The response was uploaded in the form of a supplement.

Please also note the supplement to this comment:
https://acp.copernicus.org/preprints/acp-2020-524/acp-2020-524-AC1-supplement.pdf

---

## Author Response (AR1)

First of all, we thank Paul DeMott and the anonymous Reviewer for having read the paper and for providing their careful reviews. We found the comments very helpful to improve the manuscript. Point-by-point replies to the comments are below.

For clarity and easy visualization, the Referee's comments are shown from here on in black.

> The authors' replies are in blue font with an increased indent below each of the referee's statements.

> > The relevant changes in the revised manuscript are below in green. If just a part was added to an existing sentence, then the added part is underlined. All line and page numbers in normal font refer to the unrevised manuscript. All line and page numbers **in bold** refer to the revised manuscript.

**Replies to the comments by Paul DeMott**

**General Comments**

This is a welcome and concise contribution to the literature regarding parameterization of ice nucleating particle measurements. It suggests that the size required for best relating INP concentrations to aerosols deserves revisiting in general, depending on the conditions for INP processing, and it suggests that a larger size is required than used in past parameterizations using aerosol concentration-size relations as a parameterization basis (at least for the mineral dust related parameterization they investigate). This is not an entirely new recommendation, but is clearly presented here. I do not subscribe to −15◦C as the critical or only temperature needed for constraining parameterizations as relevant to wintertime precipitation. In general, the non-comprehensiveness of using only −15◦C data is a limitation for any widespread application. I have some comments as well on clarifying the specification of size as being aerodynamic versus physical. This matters for the comparison to literature shown. In that regard, it should be made clear that the D15 parameterization is specifically for mineral dusts. The findings regarding the relation or not of apparent biological INPs to aerosol sizes is a new one to my knowledge, and is very interesting in its implications. This paper clearly motivates future studies to improve parameterizations overall through use of field data.

> We highly appreciate your general remarks regarding our manuscript. We address all concerns raised as they are listed in the specific comments below.

**Specific Comments**

1) Abstract: Lines 5-6: A key question for this might also be how divorced a surface site is from the free troposphere where clouds form? Or are the clouds always tied to the boundary layer and have cloud tops relevant to the measurements.

> We are not quite sure whether we understand this comment correctly. In principle, clouds may form above the boundary layer in air masses with a different aerosol population as compared to that found at a surface site below them. The mountain site hosting our observatory (2671 m a.s.l.) was sometimes within, sometimes above the boundary layer. As we understand, the question is whether the aerosol we sampled

at the mountain top was part of the same air mass as the clouds. Precipitation at the site typically coincided with the passage of low-pressure systems arriving from the West or Northwest and swashing across the Alps. During such a situation, the lower atmosphere is deeply mixed and we can be confident that the air at the mountain top is part of the same air mass as the precipitating clouds because sampling took place within the lower part of the clouds, as visually observed. In contrast, fair weather conditions were characterised by inversions below the sampling site. That means, the sampling site tended to be above the boundary layer and within the free troposphere. Whether it was then part of the same air mass as that of eventual clouds at greater altitude cannot be said.

We hope to have answered your question but we have not added this information to the abstract as it seems too specific for it.

2) Introduction: Line 13: To be clear, this parameterization is for mineral dust as a single category only. It should not be expected to represent a multivariate INP population.

We specified that the D15 parametrisation is for mineral dust and not expected to represent a multivariate INP population.

We rephrased the following sentences to make this clearer:

(P1L11-15; **P1L16**) A current empirical parametrisation established by DeMott et al. (2015), hereafter D15, predicts [INP] based on nucleation temperature and number concentration of mineral dust particles with diameters > 0.5 μm ($[n_{0.5}]$) (DeMott et al., 2015). Although the D15 may be applicable to temperatures below -20 °C, it is not expected to represent a multivariate INP population and remains "weakly constrained at temperatures >-20 °C, where much additional ambient and laboratory data are needed" (DeMott et al., 2015).

Lines 17-19: While this dive into the meteorology/climatology of winter precipitation is much appreciated, I want to note a caveat with regard to these studies. First, they were dominated by observations from over continental regions that were largely non-mountainous, unlike the site in the present study. Secondly, colder cloud top modes for precipitation are also noted in the referenced studies. This is consistent with earlier work by Rauber et al. (1987) from over mountainous regions in the United States, where both the −15∘C maximum associated with the dendritic growth regime and colder cloud top precipitation events are noted. There is no way to know when colder-topped clouds may be impacting the persistence of liquid water in the −15∘C regime, and thereby altering the microphysical scenario. Schultz et al. (2001) critiqued the inadequacy of this isotherm alone (in the Wetzel et al. scheme) for predicting heavy precipitation that can occur over a broad range of cloud top temperatures. So perhaps note this as "one of the modes" for winter precipitation. Hence, while it is reasonable to select this temperature for the present study, a caveat is needed because it is not possible to sit in any one location to make measurements and assume that the only relevant temperature for clouds passing over is a single value.

We recognize these studies have their limitations and agree that it is worth elaborating their findings. We added more details regarding the work done by Hanna et al. (2008). In the revised manuscript we also refer to the work by Rauber et al. (1987). Thank you for making us aware of it. The method applied in Wetzel and

Martin (2001) was indeed criticised and uses data from the United States similar to that in Hanna et al. (2008). Therefore, after a critical analysis, we concluded that Wetzel and Martin (2001) was maybe not an appropriate reference in our context. Explaining in more detail why we focus on -15 °C, we also mention in the revised version the need for investigations at other temperatures.

We reformulated and added the following sentence and text passage to the manuscript:

(P1L15-17; **P1L22-P2L24**) Cloud-top temperatures associated with winter snowfall have a primary temperature mode near −15 °C, as derived from close to $10^5$ parallel observations of cloud-top temperatures and falling solid precipitation throughout the United States (Hanna et al., 2008).

(P1L17-19; **P2L25-34**) The majority of these observations were for light snowfall. In contrast, cloud-top temperature distributions for moderate and heavy snowfall are bimodal with a second, minor mode around -40 °C (Hanna et al., 2008). This is consistent with observations in mountainous regions (Rauber et al., 1987). Of all snowfall observations with cloud-tops above homogeneous freezing temperature (i.e. >-38 °C), approximately 30% are associated with cloud-top temperature not colder than -15 °C (Hanna et al., 2008). Therefore, substantial fractions of initial ice crystals in snow-producing mixed-phase clouds may be caused by INPs that nucleate ice at temperatures ≥-15 °C (INPs$_{-15}$). This inference may extend to other midlatitude continental regions. Considering -15 °C as a temperature that is important for snow formation also makes physical sense, because maximum depositional growth of ice crystals is around -15 °C (Rogers and Yau, 1989). We, therefore, in this work, focus on INPs active at that temperature, although other temperatures would benefit from future investigations.

Line 20: What does the statement "−15∘C or warmer" mean exactly? It is repeated in the Fig. 1 caption, but the "or warmer" part is not explained or justified.

With this statement we mean INPs active at -15 °C, including all INPs that are activated at temperatures warmer than -15 °C.

We rephrased these sentences:

(P1L20; **P2L30**) […] INPs that nucleate ice at temperatures -15 °C (INPs$_{-15}$)

(Fig. 1, **Fig.3**) Cumulative concentrations of ice nucleating particles active at -15 °C [INP$_{-15}$]

Also, with regard to the statement on the sizes of INPs relevant at this temperature, I would note that widespread measurements over continents in winter are, however, limited. Measurements of size-resolved INPs are even rarer in winter, at least to my knowledge of locations and the number of measurements that have been made of INP sizes. I understand that larger sizes indicate a better correlation in this paper, but I suggest care in assuming that the role of only larger INPs in this temperature regime is confirmed as relevant to winter clouds.

As mentioned in the manuscript, three previous studies show from size-resolved measurements of INPs show that INPs$_{-15}$ are mostly > 2 µm in diameter (Huffman et al., 2013; Mason et al., 2016; Creamean et al.; 2018). These were done at different sites and seasons. The better correlation with larger sizes in this paper can be explained by the actual size of INPs$_{-15}$ found in previous studies. However, we agree that the observations are sparse, especially in winter. We added the correlation coefficients for the fits (Table 1) as well as comparisons of predicted versus measured [INP$_{-15}$] for different prediction options (Fig. S3). Furthermore, we have added the following sentence:

(P1L19; **P2L36**) Findings from a sparse number of size-resolved measurements of atmospheric INPs show that INPs$_{-15}$ are mostly > 2 µm in diameter (Huffman et al., 2013; Mason et al., 2016; Creamean et al., 2018)

(P3L85; **P6L155**) However, choosing the actual size range of INPs$_{-15}$ for the parametrisation can further improve the predictions.

(P3L82; **P5L145**) Overall, for each air mass class, the correlation coefficient of the obtained function is equal or higher with [$n_{2.0}$] as a predictor than with [$n_{0.5}$] (Table 1). This confirms that [$n_{2.0}$] is a more powerful predictor of INPs$_{-15}$ than [$n_{0.5}$] when combined with air mass differentiation (Fig. S3). It underlines the importance to consider aerosol particles > 2 µm. To further develop a parametrisation valid for temperatures >-20 °C, we suggest to further investigate the presented functions, because INPs active at other temperatures or at other locations and during different seasons may also be associated with other particle sizes or other INP concentrations. Especially the addition of INPs in precipitating air masses should be constrained with data from all over the globe.

[Figure]

(Fig. 1; **Fig. 3**) Power functions for each type of air masses based on [$n_{0.5}$] and [$n_{2.0}$] are shown. An additional preliminary parametrisation for precipitating air masses based on [$n_{2.0}$] is shown (PRECIP+, thick dark blue line). It is the same as for non-precipitating air masses but with an added constant equivalent to 0.014 INPs L$^{-1}$. The corresponding equations and R$^2$ values are shown in Table 1.

(**Table 1**)

**Table 1.** Equations of the functions shown in Fig. 3 (i.e. PRECIP, PRECIP+, NON-PRECIP, SD) predicting cumulative concentrations of ice nucleating particles active at -15 °C [INP$_{-15}$] based on aerosol particles with aerodynamic diameters > 0.5 µm [$n_{0.5}$] and > 2 µm [$n_{2.0}$] and their respective $R^2$ values. In addition, equations and $R^2$ values of power functions fitted to all data points irrespective of air mass classes are shown (ALL). The equations are listed based on the following formula: $y = b * x^a + c$, with y equal to [INP$_{-15}$].

| Air mass type | n | $x$ | $b$ | $a$ | $c$ | $R^2$ |
|---|---|---|---|---|---|---|
| ALL | 124 | [$n_{0.5}$] | 0.02 | 0.19 | 0 | 0.06 |
| PRECIP | 56 | [$n_{0.5}$] | 0.03 | 0.23 | 0 | 0.05 |
| NON-PRECIP | 57 | [$n_{0.5}$] | 0.01 | 0.55 | 0 | 0.29 |
| SD | 11 | [$n_{0.5}$] | 0.001 | 1.34 | 0 | 0.55 |
| | | | | | | |
| ALL | 124 | [$n_{2.0}$] | 0.03 | 0.22 | 0 | 0.07 |
| PRECIP | 56 | [$n_{2.0}$] | 0.09 | 0.36 | 0 | 0.12 |
| PRECIP+ | 56 | [$n_{2.0}$] | 0.58 | 1.19 | 0.014 | 0.14 |
| NON-PRECIP | 57 | [$n_{2.0}$] | 0.58 | 1.19 | 0 | 0.44 |
| SD | 11 | [$n_{2.0}$] | 0.02 | 0.99 | 0 | 0.55 |

(**Fig. S3**)

[Figure]

**Figure S3.** Measured and predicted [INP$_{-15}$] (std L$^{-1}$) for (**a**) the D15 parametrisation, (**b**) prediction based on a single trendline fitted through all data of aerosol particles with aerodynamic diameters > 0.5 µm [n$_{0.5}$], (**c**) predictions based on [n$_{0.5}$] and three different trendlines fitted through the data of PRECIP (blue circles), NON-PRECIP (green triangles), and SD (red squares) air masses, and (**d**) same as (c), but based on aerosol particles with aerodynamic diameters > 2.0 µm [n$_{2.0}$]. Shapes of symbols in (a) and (b) are consistent with those in (c). However, they are coloured in gray as the prediction is independent of air mass classes. A range of a factor of two (dotted lines) about the 1:1 line (solid line) as well as the percentage of values lying within that range are shown in all panels.

**3) Material and Method**

Line 33: Has the Coriolis collection method been compared to any other standard method, such as filters? This would be good to know.

> We are not aware of a study on atmospheric INPs in which the Coriolis®µ and other samplers were operated in parallel for direct comparison. Other samplers we have in our lab operate with about 20 times lower flow rates and require operating for several hours before a meaningful analysis of INPs$_{-15}$ is possible, while the Coriolis needs to be operated for 10 or 20 minutes only. Therefore, a direct comparison is not that

straight forward. Nevertheless, [INP] collected for 20 min with the Coriolis®µ was compared to 3h and 24h filter (pore sizes 1 µm) samples on different days and sampling intervals and found comparable INP trends for both sampling methods (Tarn et al., 2020).

We have added the following sentence:

(P2L34; **P3L65**) Aerosol samples were collected using the Coriolis®µ as was done in recent studies (Els et al., 2019; Tarn et al., 2020; Miller et al., 2020).

Line 36: Can you clarify why there is an upper bound to measured concentrations?

Yes, we can clarify this issue:

(P2L36; **P3L72**) Sampling and analysis were designed in such a way that expected [INP$_{-15}$] of each sample was well within the detection limits, meaning that several but not all droplets in the assay would be frozen. With our sampling and analysis design the detection range lies between $4.8 \times 10^{-4}$ (i.e. first drop frozen) and $8.1 \times 10^{-2}$ L$^{-1}$ (i.e. second last drop frozen). In 15 samples, all droplets were frozen and in one sample no droplet was frozen at -15 °C. These samples were not considered because their [INP$_{-15}$] were outside the detection limits. For the other samples (n = 124) several, but not all droplets froze.

Line 37: Does the detection limit mean the stated lower limit of quantification or what does it mean? I assume the background comparison would use particles per ml, rather than per L of air. Can you state values in that manner? Does it literally mean that no wells froze when you removed ultrapure water from the sampler and ran the same cooling rates?

Exactly, it means that no wells froze, when we removed ultrapure water from the sampler and ran the drop freezing experiment at the same cooling rate. By specifying above what we mean by the detection limit, we hope to have clarified the issue.

Line 39: I am curious about the use of specific APS channel bin limits to define [n$_{0.5}$]and [n$_{2.0}$]. Aerodynamic size is different from physical size, and if using APS data to constrain values from a parameterization, especially D15, it requires conversion first to equivalent spherical physical diameter (using effective density or density and an assumed shape factor), and then recalculation of the concentrations above 500 nm diameter. Otherwise the concentrations could be biased high for use in D15.

The specific APS channel bin limits were already defined in the manuscript (P2L39-40) and is ≥ 0.542 µm and ≥ 1.982 µm. However, we clarified the scanning properties of the APS, added the D15 equation, mentioned that we used the aerodynamic size and discussed the consequences of using aerodynamic size instead of the physical size. The conversion to physical diameter is not possible for multivariate aerosol populations without making assumptions, since we have no direct evidence at hand regarding the actual shapes and densities of the probably large variety of sampled particles.

The following sentences were rephrased or added:

(P2L37-40; **P3L78**) Number concentrations of particles [n] were measured from 0.5 µm to 20 µm (51 bins) with 20 s scanning time with the APS […]

(P2L40; **P3L82**) [INPs-15] estimates based on D15 were calculated as:

$$INP_T = cf * n_{0.5}{}^\beta * e^{\gamma*(-T)+\delta},$$

where *cf* = 1, $\beta$ = 1.25 , $\gamma$ = 0.46, $\delta$ = -11.6, T is the temperature in degree Celsius, $n_{0.5}$ is the aerosol concentration with diameter ≥ 0.542 µm (std $cm^{-3}$), and $INP_T$ the ice nucleation particle concentration (std $L^{-1}$) at T in Celsius.

(P2L40; **P3L88**) The aerodynamic particle diameters as determined by the APS are not the same as the physical diameters on which D15 is based. To transform the APS data into physical diameters would require information, which is not available, about densities and shapes of the main components of sampled particle populations. If actual particle densities were mostly > 1 g $cm^{-2}$, our [$n_{0.5}$] would be somewhat higher than if they would have been calculated as physical particle diameters.

Line 46: Can you clarify what is meant by "amongst others" used here.

Yes, we clarified this.

(P2L46; **P4L98**) Along the trajectories, total precipitation was traced amongst others (i.e. height, pressure, temperature, specific humidity and surface height) enabling us to determine the total precipitation amount along the last 6 hours prior to sampling (Fig. S2).

4) Results and Discussion

Line 59: As mentioned already, please clarify how [$n_{0.5}$] as defined by the APS is used in D15.  Also, I infer from Fig. 1 that you have not used the scaling factor derived in D15 for application in modeling total INP concentrations, but you have not stated that. It means that you do not assume the correction of continuous flow diffusion chamber data that was demonstrated valid for laboratory and transported ambient Saharan dust in D15. Please say so and perhaps justify.

As mentioned already above, we clarified how we calculated [INP] estimates based on D15. As you inferred, we did not use a scaling factor to derive D15. Furthermore, we now mention this in the revised version of the manuscript and add a justification for keeping the scaling factor equal one:

(P2L40; **P3L86**) No calibration factor was necessary (cf = 1) because INPs were observed in immersion mode (via a drop freezing assay) and not for instance, in a continuous flow diffusion chamber, where, because of relative humidities below 100%, only part of the INPs passing the instrument may become immersed in liquid droplets.

Note on Figure 1: The listing of correlation coefficients for the fits shown, and perhaps also for the 500 nm concentrations (could list all of these in a table, for example), would be useful.  This would put some quantification into the statement about "clarity of separation" of data when 2 microns is used as the reference. Also, the diamonds in the figure are extremely hard to resolve, but perhaps at a full page size the figure will be easier to read. Finally, please clarify again in the caption if the x-axis concentration values are for aerodynamic or physical size.

We added power functions for 0.5 µm concentrations in the figure and we listed the equations and their $R^2$ values in a table (Table 1). Additionally, we made the plots more reader friendly and divided Fig. 1 into four figures (Fig. 1-4).

We made the following changes:

(Fig. 1a-c; **Fig. 1**)

[Figure]

**Figure 1.** (a) Five-days and (b) six-hours back trajectories of air masses that were precipitating (PRECIP, blue), non-precipitating (NONPRECIP, green), and carrying a substantial fraction of Saharan dust while non-precipitating (SD, red).

(Fig. 1d-f; **Fig. 2**)

[Figure]

**Figure 2.** Number concentrations of aerosol particles with aerodynamic diameters (a) > 0.5 µm [$n_{0.5}$] and (b) > 2.0 µm [$n_{2.0}$] and (c) their ratio for the aerosol populations of PRECIP, NON-PRECIP and SD air masses.

(Fig. 1g-h; **Fig. 3**)

[Figure]

**Figure 3.** Cumulative concentrations of ice nucleating particles active at −15 °C [INP$_{-15}$] (a) versus [$n_{0.5}$] and (b) versus [$n_{2.0}$] for PRECIP (blue circles), NON-PRECIP (green triangles), and SD (red squares) air masses. The D15 parametrisation extrapolated to −15 °C is shown as a black, dotted line. Power functions (solid lines) for each type of air masses based on [$n_{0.5}$] and [$n_{2.0}$] are shown. A preliminary parametrisation for precipitating air masses based on [$n_{2.0}$] is shown (PRECIP+, thick dark blue line). It is the same as for non-precipitating air masses but with an added constant equivalent to 0.014 INPs L$^{-1}$. The corresponding equations and R$^2$ values are shown in Table 1.

(Fig. 1i; **Fig. 4**)

[Figure]

**Figure 4.** Difference between [INP$_{-15}$] in precipitating (PRECIP) air masses and the function fitted to the non-precipitating (NON-PRECIP) air masses (additional [INP$_{-15}$] in PRECIP) versus [$n_{2.0}$]. The median difference is 0.014 std L$^{-1}$ (black dotted line) and the lower and upper quartiles are 0.003 and 0.047 std L$^{-1}$, respectively (grey dashed lines). The linear fit (grey solid line) is weakly positive but not significant (Pearson correlation test, R = 0.0027 and p = 0.98). Circle area is proportional to the amount of precipitation along the last 6 hours of the trajectory prior to sampling. Black circles are for samples precipitating at Weissfluhjoch.

(P3L82; **P5L145**) Overall, for each air mass class, the correlation coefficient of the obtained functions is equal or higher with [$n_{2.0}$] as a predictor than with

[$n_{2.0}$] (Table 1). This confirms that [$n_{2.0}$] is a more powerful predictor of INPs$_{-15}$ than [$n_{0.5}$] when combined with air mass differentiation (Fig. S3).

(**Table 1**)

**Table 1.** Equations of the functions shown in Fig. 3 (i.e. PRECIP, PRECIP+, NON-PRECIP, SD) predicting cumulative concentrations of ice nucleating particles active at -15 °C [INP$_{-15}$] based on aerosol particles with aerodynamic diameters > 0.5 μm [$n_{0.5}$] and > 2 μm [$n_{2.0}$] and their respective $R^2$ values. In addition, equations and $R^2$ values of power functions fitted to all data points irrespective of air mass classes are shown (ALL). The equations are listed based on the following formula: $y = b * x^a + c$, with y equal to [INP$_{-15}$].

| Air mass type | n | $x$ | $b$ | $a$ | $c$ | $R^2$ |
|---|---|---|---|---|---|---|
| ALL | 124 | [$n_{0.5}$] | 0.02 | 0.19 | 0 | 0.06 |
| PRECIP | 56 | [$n_{0.5}$] | 0.03 | 0.23 | 0 | 0.05 |
| NON-PRECIP | 57 | [$n_{0.5}$] | 0.01 | 0.55 | 0 | 0.29 |
| SD | 11 | [$n_{0.5}$] | 0.001 | 1.34 | 0 | 0.55 |
| | | | | | | |
| ALL | 124 | [$n_{2.0}$] | 0.03 | 0.22 | 0 | 0.07 |
| PRECIP | 56 | [$n_{2.0}$] | 0.09 | 0.36 | 0 | 0.12 |
| PRECIP+ | 56 | [$n_{2.0}$] | 0.58 | 1.19 | 0.014 | 0.14 |
| NON-PRECIP | 57 | [$n_{2.0}$] | 0.58 | 1.19 | 0 | 0.44 |
| SD | 11 | [$n_{2.0}$] | 0.02 | 0.99 | 0 | 0.55 |

Line 68-69:  Although I infer that you are getting at the enrichment of INPs that are inferred to be of biological origin following rain events, the statement "enrichment of the aerosol population with highly efficient INPs during precipitation" was a little confusing, because your data show that aerosol concentrations under rain events are typically lower on average.

Yes, we think that precipitation adds highly active INPs as aerosols which are not changing the overall aerosol concentration significantly. Probably, precipitation washes out a part of the background aerosol particles from the atmosphere, while at the same time aerosolising a smaller number of aerosol particles. If latter contains a much higher fraction of INPs than present in the washed-out air mass, the overall effect will be a smaller aerosol number concentration containing an increased INP concentration. We changed our wording as the following:

(P3L68; **P5L133**) In precipitating air masses, the ratio between [INP$_{-15}$] and [$n_{2.0}$] was usually larger than in non-precipitating air masses. This reveals that the aerosol population was enriched with INPs active at moderate supercooling during precipitation, consistent with previous findings (Bigg and Miles, 1964; Huffman et al., 2013).

Line 70: suggest "masses" for "mass"

We rephrased it into: "history of air masses"

Lines 73-75: While you have used a constant here to describe an assumed physical process whereby the production is independent of aerosols already in the air, I am curious if there is any dependence of this process on precipitation rate? Or in a complex way on that and existing surface wetness? Does it always produce large INPs?

> Thank you for this question. Indeed, it is very interesting to further understand the assumed physical process. We added a sentence in the introduction and a small analysis of the most common land covers these trajectories have passed.
>
> > (P1L22; **P2L41**) This might be explained by aerosolisation of INPs by rain itself, a mechanism similar to the generation of bioaerosol by raindrop impingement (Joung et al., 2017), which is probably dependent on various parameters like surface wetness or land cover.

Line 90: As stated earlier, I do not feel that such a strong statement can be made regarding the importance primarily of the −15 °C INPs. Not without specific inspection for many sites around the world. And I believe that literature supports that this single temperature is not exclusive for predicting wintertime precipitation.

> We clearly recognize the limitations of our focus on one temperature and changed the wording in the Conclusions accordingly:
>
> > (P4L89-91; **P6K164**) While a new parametrisation for a previously weakly constrained temperature is clearly a step forward, it complements rather than replaces previous parametrisations.

> Note that, references are at the end of this document.
* * *
**Authors' response to anonymous Referee #2**

General comments: This reviewer enjoyed reading this concise paper. The study presents several important aspects, such as precipitation INPs & their source appointment & implication of supermicorn INPs, that all could be an important addition to ACP. While the manuscript is overall well written, some sections deserve more articulations in this reviewer's opinion. See specific/technical comments given below. The reviewer hopes these help the authors and improve the manuscript.

> We highly appreciate your general remarks regarding our manuscript. Your comments were very helpful to us in improving the manuscript.

The reviewer sees some speculative sentences (e.g., P3L71-75, P3L80-82). Concerning these, at the end, claims of novelty and priority (P3L85-P4L91) are questionable to this reviewer.

If we understand correctly, the problem is with part of the Conclusion (P3L85-91) apparently being based on speculation (P3L71-75, P3L80-82). The speculation (P3L71-75, i.e. that additional [INP] in precipitating air masses is independent from [n]) is indeed critical for the mentioned part of the Conclusion. However, it was not left standing as a speculation but was confirmed by the subsequently presented results (P3L75-80, i.e. the relationship between additional INP in precipitating air masses and [n] is not significant). From our point of view, these results justify the Conclusion. We have restructured the Results and Discussion and mention the possible explanation for aerosolization of INPs by rain (P3L71-73) in the introduction. The second speculation (P3L80-82) remains as such and is merely an interpretation of the first, confirmed speculation. The second speculation is not reiterated as a conclusion. We removed the speculation about the additional INPs possibly being fungal spores. We calculated the mean length of the trajectories 6 h prior sampling and determined the relative proportions of countries and land covers associated with the trajectories.

(PL23; **P2L40**) Furthermore, an increase in atmospheric abundance of INPs active at moderate supercooling temperatures, such as -15 °C, have been observed during precipitation (Biggs and Miles, 1964; Huffman et al., 2013; Hara et al., 2016; Conen et al., 2017). This might be explained by aerosolisation of INPs by rain itself, a mechanism similar to the generation of bioaerosol by raindrop impingement (Joung et al., 2017), which is probably dependent on various parameters like surface wetness or land cover.

(**P4L111**) Six hours before arriving at Weissfluhjoch, the trajectories crossed a mean distance of 242 (s.d.±145) km and spent two third of the time over Switzerland (Fig. 1b). Forested land (31%), agricultural fields (17%), pasture (12%) and natural grasslands (12%) were the most common land covers they were passing, as derived by the European Copernicus programs Corine land cover map (https://land.copernicus.eu/pan-european/corine-land-cover/clc2018).

(P3L64-80; **P5L126**) Considering the fact that the observed size of $INPs_{-15}$ is mostly larger than 2 µm (Huffman et al., 2013; Mason et al., 2016; Creamean et al., 2018), we plot measured $[INP_{-15}]$ against $[n_{2.0}]$, instead of $[n_{0.5}]$, resulting in a more distinct separation of the data to the different air masses (Fig. 3b). In each of the three categories of air masses, $[INP_{-15}]$ can be described as a function of $[n_{2.0}]$ that is valid for a range of $[INP_{-15}]$ from 0.0006 std $L^{-1}$ to 0.14 std $L^{-1}$ (Table 1). For data in SD and non-precipitating air masses, $[INP_{-15}]$ can be described as power functions of $[n_{2.0}]$ with similar linear slopes on a log-log plot, but lower $[INP_{-15}]$ per $[n_{2.0}]$ for SD. This goes hand in hand with the earlier observations that air masses influenced by SD carry less INPs active at moderate supercooling per unit mass of aerosol particles than European background air masses (Conen et al., 2015). In precipitating air masses, the

ratio between $[INP_{-15}]$ and $[n_{2.0}]$ was usually higher than in non-precipitating air masses. This reveals that the aerosol population was enriched with INPs active at moderate supercooling during precipitation, consistent with previous findings (Bigg and Miles, 1964; Huffman et al., 2013). Since additional INPs during precipitation might be due to aerosolisation of INPs by rain which is likely independent of the background in $[n]$, we describe $[INP_{-15}]$ in precipitating air masses by adding a constant to the function fitted to the non-precipitating cases (Fig. 3b). The median difference between the function of nonprecipitating air masses and measured $[INP_{-15}]$ in precipitating air masses was 0.014 std $L^{-1}$ (Fig. 4). The relationship between these differences and $[n_{2.0}]$ was weakly positive and not significant, meaning that the absolute value of additional INPs in precipitating air masses was independent of $[n_{2.0}]$. This finding corroborates our assumption that additional INPs during precipitating air masses are independent of background $[n]$. A consequence of our finding is that to precipitating air masses with low $[n_{2.0}]$, the addition of INPs aerosolised by precipitation makes a relatively large contribution to the overall $[INP_{-15}]$. The additional INPs during precipitation could be emitted through the impact of rain on snow-free lower lying plain regions, a speculation which needs to be investigated in future.

The reviewer understands that the parameterization of cumulative INPs @ -15 dC for 1.5 months is a snapshot example of the authors' concept/proposal of IN parameterization for moderate spercooling. Regardless, an elaborated summary on how specifically beneficial the proposed parameterization would be to the community seem necessary in the conclusion section.

It is indeed a concept. Hence, the title begins with "Towards...". We think that our proposed parametrisation is beneficial to further investigate the role of INPs on precipitation, especially in a changing climate, which includes changes in precipitation patterns. Accordingly, we have extended the Conclusion section:

(P3L84-91; **P5L153**) In summary, it is possible to reconcile two fundamental aspects of INPs active at moderate supercooling - increased abundance during precipitation and size - with a widely used approach to parametrise INPs active at colder temperatures. However, choosing the actual size range of INPs$_{-15}$ for the parametrisation can further improve the predictions. An even greater improvement in predictions is possible when we additionally distinguish between air masses that are precipitating, non-precipitating, or carrying a substantial fraction of Saharan dust. More of the variance of INP concentrations was explained by aerosol concentrations in air masses that were non-precipitating or carrying desert dust as compared to air masses that were precipitating. The absolute value of additional INPs in precipitating air masses seems to be independent of total aerosol concentrations.
To tackle predictions of INPs active at moderate supercooling, particular attention has to be attributed to sample larger aerosol particles at mixed-phase cloud height, including air masses that have been precipitating, and

, as was done in this study. Future observations should constrain the presented parametrisation functions in a wider geographical context. While a new parametrisation for a previously weakly constrained temperature is clearly beneficial, it complements rather than replaces previous parametrisations. In a changing climate, with increasing temperatures and changing precipitation patterns, it is important to predict feedbacks between INPs and precipitation.

The authors may want to soften the tone regarding the comparison to D15 etc., and carefully re-formulate all the associated statements. The take-home message of the paper should not be 'a previous parameterization does not work' (this reviewer felt this to some extent). Instead, the authors should structure the discussion by blending/merging previous contributions in this work (rather than clipping) and emphasize how their new parametrization can complement the previously introduced one(s). Plus, the reviewer questions how meaningful INP measurements at single T (-15C) would be concerning many speculations and absurd statements given in the manuscript. The reviewer is aware that the justification of this single T point is given in P1L15-17. Nonetheless, the authors need to soften their tones in some sections.

> Thank you bringing this up. It was not our intention to leave the reader with such a take-home message. We have carefully reformulated the statements associated with the comparison to D15 and other parametrisations. Furthermore, we recognized the limitations of our parametrisation and mentioned that investigations of the parametrization of further temperatures are needed. We added or changed the following sentences:
>
> > (P1L11; **P1L14**) The past decade has seen substantial efforts toward improving empirical parametrisations of [INP] (e.g., DeMott et al., 2010; Phillips et al., 2013; DeMott et al., 2015).
> >
> > (P1L21; **P2L38**) This particle size is however under-represented for instrumental reasons in the empirical data on which D15 and other parametrisations (e.g., DeMott et al., 2010; Phillips et al., 2013) are based.
> >
> > (P3L85; **P5L154**) Parametrisations based on the number concentration of aerosol particles are reasonable to predict INPs at moderate supercooling.
> >
> > (P4L91; **P6L164**) While a new parametrisation for a previously weakly constrained temperature is clearly beneficial, it complements rather than replaces previous parametrisation.
>
> Furthermore, we elaborated on the justification of - 15°C, as the following:
>
> > (P1L15-17; **P1L21**) The coldest part of a cloud – typically cloud-tops and their temperature – determines what fraction of the INP population will get activated and form ice crystals. Any INPs with colder activation temperature will remain inactive. Cloud-top temperatures associated with winter snowfall

have a primary temperature mode near −15 °C, as derived from close to $10^5$ parallel observations of cloud-top temperatures and falling solid precipitation throughout the United States (Hanna et al., 2008). The majority of these observations were for light snowfall. In contrast, cloud-top temperature distributions for moderate and heavy snowfall are bimodal with a second, minor mode around -40 °C (Hanna et al., 2008). This is consistent with observations in mountainous regions (Rauber et al., 1987). Of all snowfall observations with cloud-tops above homogeneous freezing temperature (i.e. >-38 °C), approximately 30% are associated with cloud-top temperature not colder than -15 °C (Hanna et al., 2008). Therefore, substantial fractions of initial ice crystals in snow-producing mixed-phase clouds may be caused by INPs that nucleate ice at temperatures ≥ -15 °C (INPs$_{-15}$). This inference may extend to other midlatitude continental regions. Considering -15 °C as a temperature that is important for snow formation also makes physical sense, because maximum depositional growth of ice crystals is around -15 °C (Rogers and Yau, 1989). We, therefore, in this work, focus on INPs active at that temperature, although other temperatures would benefit from future investigations.

(P1L19; **P2L34**) Based on current understanding, atmospheric INPs$_{-15}$ are mostly biological aerosol particles (Murray et al., 2012). Although their number concentration is often small, primary ice by INPs$_{-15}$ may get multiplied by an order of magnitude due to secondary ice formation (Mignani et al., 2019).

Specific and technical comments:
P1L2: –> …at -15 C, at which a cloud-top…

O.k.

(P1L1-3; **P1L2**) Only a minute fraction of aerosol particles, so-called ice nucleating particles (INPs), can trigger initial ice formation at -15 °C, at which cloud-top temperatures are frequently associated with snowfall.

P1L3-4: –> We found at a mountain top site in Swiss Alps that INPs active at -15 C contained different fraction of coarse (>2 um) aerosol particles, depending on…

The suggested reformulation would change the meaning of our sentence, therefore we decided to keep our phrasing.

P1L10-11: If the authors are discussing about INPs in general (not droplet freezing in particular), then the water saturation should be one of the most important factors/variables to complicate our understanding in [INP] and should be included in this particular sentence. Plus, there should be more comprehensive and appropriate references to cite and support this sentence other than what appears here.

Thank you for the comment. We agree that this sentence was not broad enough to generally introduce INPs. To start with, we added a sentence about the importance of

ice in clouds. Furthermore, we changed the particular sentence and referred to multiple other papers to give an overview of the challenges in INP observations. Water saturation is one of the challenges to detect INPs using Continuous Flow Diffusion Chambers or similar instruments. Although it is an important variable, it falls out of the scope of this study where INPs are only determined in immersion mode.

(P1L9; **P1L10**) The presence of ice in clouds is important for precipitation initiation (Mülmenstädt et al., 2015; Heymsfield et al., 2020).

(P1L10; **P1L12**) The difficulty of understanding and thus predicting atmospheric INP concentrations ([INP]) originates from observational challenges related to field measurement techniques (Cziczo et al., 2017), the large variety of potential sources (Kanji et al., 2017), and the wide range in atmospheric abundances from $10^{-6}$ to $10^3$ $L^{-1}$ (Petters and Wright, 2015).

P1L13-14: Although successful… –> Although the D15 parameterisation may be applicable to temperatures below -20 C, it remains…

O.k.

P1L18: RE "it makes physically sense for various reasons" – then, the reviewer suggests the authors to briefly, at the least, discuss physical mechanisms of why it peaks specifically at -15 C for the readers. Otherwise, -15 sounds like a magic number, making the manuscript and its concept sound speculative, in the reviewer's opinion.

We rephrased this sentence and give a reason why -15 °C makes sense to consider.

(P1L18, **P2L30**) This inference may extend to other midlatitude continental regions. Considering -15 °C as a temperature that is important for snow formation also makes physical sense, because maximum depositional growth of ice crystals is around -15 °C (Rogers and Yau, 1989). We, therefore, in this work, focus on INPs active at that temperature, although other temperatures would benefit from future investigations.

P1L19-21: –> Recent findings from…; Creamean et al., 2015). However, a particle size is poorly represented in the empirical data-based parameterizations (Phillips et al., 2013). Perhaps, this part reads better this way?

Yes, thank you, we reformulated this sentence according to your suggestion.

P1L22: –> such high temperature active INPs

We reformulated it in the following way:

(P1L22; **P2L40**) Furthermore, an increase in atmospheric abundance of INPs active at moderate supercooling have been observed during precipitation

(Bigg and Miles, 1964; Huffman et al., 2013; Hara et al., 2016; Conen et al., 2017).

Section 2: How turbulent was it while sampling during snow precipitation? Some discussion of local meteorology and its impact on sampling/measurements beyond crude categorization based on observation seem necessary.

Daily averages of basic meteorological parameters provided by MeteoSwiss measured at the site are provided in the supplement (Fig. S1). Also, we have added (i) a sentence that we cannot rule out resuspension of ice particles from the surface, but that it is unlikely that the resuspended particles contribute significant amounts to the total sampled particles, and (ii) added the local wind speeds during sampling intervals, which were 7.1 (s.d.±3.4) m s$^{-1}$. Please note that the categorization is not crude but based on trajectories calculated with LAGRANTO that was driven by wind fields of the Swiss National Weather Service and the European Centre for Mid-Range Weather Forecast, as was described in the method section. More details of the precipitation along the trajectories for each category is provided in the supplement as well (Fig. S2).

(**Fig. S1**)

[Figure]

**Figure S1.** Daily averaged meteorological data at Weissfluhjoch during "Role of Aerosols and Clouds Enhanced by Topography on Snow (RACLETS)" campaign, including air temperature (T, °C), relative humidity (RH, %), pressure (p, hPa), wind speed (ws, m s$^{-1}$), wind direction (wd, °), and precipitation rate (mm h$^{-1}$). Precipitation data were missing prior to 25 February, therefore those of the station in Davos (DAV) are shown as well (gray dots). Number of impinger-based aerosol samples with quantified [INP$_{-15}$] (n = 124) for air masses that were non-precipitating (green triangles, n = 57), precipitating (blue circles, n = 56), and carrying a substantial fraction of Saharan dust while non-precipitating (red squares, n = 11) are shown.

(P2L30; **P3L60**) Ice particles resuspended from surrounding surfaces (snow-covered throughout the campaign and with average local wind speed of 7.1 (s.d.±3.4) m s$^{-1}$ during sampling intervals) cannot be ruled out, but are unlikely to contribute significant amounts to the total sampled particles.

(**Fig. S2**)

[Figure]

**Figure S2.** Total precipitation along the last 6 hours (mm) of the trajectory prior to sampling for air masses that were precipitating (PRECIP, blue), non-precipitating (NON-PRECIP, green), and carrying a substantial fraction of Saharan dust while non-precipitating (SD, red). Histograms of 0.5 mm binned values are plotted (dodged). The mean of each category is denoted. The precipitation values were derived along the LAGRNTO backward trajectories.

P2L26-27: –> While the site, surrounding mountains and nearby valleys were snow-covered, most of the foothill regions were not, and only rain persisted in those regions during our study period. Is this what the authors meant?

This is partly what we meant. More specifically, we meant that most of the lower lying plain and the foothill regions were not snow-covered, and that precipitation occurred in form of rain in those regions during our study period. Note that the 6h backwardtrajectories extend well into the Swiss Plain, and other plain regions. We calculated the mean length of the trajectories 6h prior sampling and determined the most frequent countries and land covers they travelled over.

We reformulated the sentence accordingly:

> (P2L26-27; **P2L48**) The site, surrounding mountains and nearby valleys were snow-covered, while most of the lower lying plain and the foothill regions were not, and precipitation occurred in the form of rain in those regions during our study period.

P2L31: The reviewer requests the authors to describe their inlet configuration in the manuscript. Was it a TSP isokinetic inlet? What was the height of the top of the inlet? Particle loss/transmission well characterized? How did the authors make sure there is no re-suspension of snow or soil got into the inlet as well as no influence of local gust/turbulence and other dynamic/thermodynamic effects?

Thank you for your questions related to the inlet. As requested, we have added a detailed description of the inlet to the method section, which hopefully answers your questions.

We added the following paragraph to the manuscript:

> (P2L31; **P2L53**) Total aerosol was sampled through a heated inlet (heating element kept at +46 °C) similar to the one described in Weingartner et al. (1999), which was designed such that particles with diameters < 40 μm are sampled up to a wind speed of 20 m s$^{-1}$. The inlet extended through the eastern wall of the laboratory and was about 8 m above local ground. The aluminium inlet tubing had an inner diameter of 4.5 cm throughout its total length of 7 m. Particles entering the inlet travelled at a speed of about 3 m s$^{-1}$ first 2.5 m downward, then turned by 70° in a radius of 20 cm towards the inside of the laboratory and continued for another 4.5 m about 20° downslope before being trapped in the impinger, approximately 2.2 s after they had entered the inlet. Ice particles resuspended from surrounding surfaces (snow-covered throughout the campaign and average local wind speed of 7.1 (s.d.±3.4) m s$^{-1}$ during sampling intervals) cannot be ruled out, but are unlikely to contribute significant amounts to the total sampled particles. The air flow was maintained throughout the campaign at 300 L min$^{-1}$, during sampling by a high flow-rate impinger (Bertin Technologies, Coriolis®μ) and between sampling intervals by a makeup flow using an external blower. In addition, an Aerodynamic Particle Sizer (APS; Model 3321, TSI Corporation) sampled from the same inlet upstream of the impinger at 1 L min$^{-1}$.

P2L32: The reviewer requests the authors to elaborate their impinger particle caption efficiency in the manuscript. To the reviewer's knowledge, an impinger is good at capturing relatively large size particles, but not that efficient on trapping small particles. How did this

kind of size-dependent trapping efficiency potentially affect the sampling activity and overall results should be addressed in the text.

The instrument is designed to collect biological particles and is constructed to sample all particles > 0.5 μm with efficiencies > 50%. In a personal communication, the manufacturer provides us with theoretical sampling efficiencies for different particle sizes > 0.5 μm, which we now included in the manuscript. How the efficiencies potentially affect the overall results is addressed in the Results and Discussion. We have added the following sentences:

(P2L32; **P3L67**) With increasing particle size the theoretical sampling efficiency of the Coriolis®μ increases from around 50% for particles of 0.5 μm in size, 80% for particles of 2 μm, to close to 100% for particles of 10 μm (personal communication with Bertin Technologies).

(P3L56; **P4L117**) Therefore, relative differences in measured [INP$_{-15}$] between precipitating and non-precipitating air masses would be affected very little, if a substantial fraction of INPs$_{-15}$ would have been of a size near 0.5 μm, which was sampled with a lower efficiency (50%) than 2 μm (80%). However, [INP$_{-15}$] in both of these air masses would have been underestimated relative to [INP$_{-15}$] in SD affected air masses, which had the highest [$n_{2.0}$] to [$n_{0.5}$] ratio.

P2L33: Perhaps some water were sucked up by a pump rather than being evaporated? Then, replenishing pure water may have affected C_INP and n_INP estimations at the end? The authors presume only water evaporated and all aerosol particles remained in an impinger jar throughout individual samplings? Would it be really the case for the impinger, which was used in this study?

Aerosol losses due to different reasons cannot be ruled out. Water losses due to evaporation can however be compensated. We changed our formulation.

(P2L34; **P3L69**) Water losses due to evaporation were compensated by replenishing the circulating water after 10 and 20 min.

P2L36: Please clarify what "15 above, 1 below range" means.

We added a few sentences regarding the detection limit in order to clarify this.

(P2L35-36; **P3L72**) Sampling and analysis were designed in such a way that expected [INP$_{-15}$] of each sample would be well within the detection limits, meaning that several but not all droplets in the assay would be frozen. With our sampling and analysis design the detection range lies between $4.8 \times 10^{-4}$ (i.e. first drop frozen) and $8.1 \times 10^{-2}$ L$^{-1}$ (i.e. second last drop frozen). In 15 samples, all dropets were frozen and in one sample no droplet was frozen at -15 °C. These samples were not considered because their [INP$_{-15}$] were outside the detection limits. For the other samples (n = 124) several, but not all droplets froze.

P2L38: APS only appears once… No abbreviation seems needed.

> True for the initial version. In the revised manuscript we use it more and therefore, we keep the abbreviation.

P2L37-39: Number concentrations of particles should have been integrated rather than being averaged, correct?

> Number concentrations of particles were calculated the following way: first we calculated the sum of the concentrations of the bins of interest for each time point of measurement (20 seconds scanning time), then we averaged these concentrations over the 20 minutes of impinger-based sampling time. We have clarified this in the following way:

>> (P2L39; **P3L78**) Number concentrations of particles [n] were measured from 0.5 µm to 20 µm (51 bins) with 20 s scanning time with the APS, were integrated (summed) from the particles sizes of interest onward (i.e. ≥ 0.542 µm for $[n_{0.5}]$ and ≥ 1.982 µm for $[n_{2.0}]$) and were averaged over each time-period (20 min) of the taken impinger-based aerosol samples.

P2L40: [n] and [INP] also scaled to standard T (273.15 K)???

> For the scaling to standard temperature, we would need to know the air temperature in the inlet at the entrance of the instrument during the collection for every aerosol sample. However, we measured the temperature of the air in the inlet only to make sure that the heated inlet works (the temperature of air at the end of the inlet (the entrance of the Coriolis) during this check was +16 °C). Therefore [n] and [INP] is scaled to standard pressure only, as we have mentioned. We now defined the abbreviation "std" more obviously. Note, however, that the change in the concentrations due to scaling to standard T would change the concentrations by less than 5%. This is low compared to the scaling to pressure and negligible compared to the error in measurement.

>> (P2L40; **P3L81**) $[n_{0.5}]$, $[n_{2.0}]$ and [INP] were adjusted to standard pressure conditions (std; $P_{ref}$ = 1013.25 hPa).

P2L49: aerosol samples –> impinger samples

> The term "aerosols" relates to aerosol particles immersed into the air. Our samples contain aerosols. Therefore, we would prefer to keep the term aerosol samples throughout the manuscript and just mention "impinger-based aerosol samples" specifically:

>> (P2L49; **P4L103**) From the total of 124 impinger-based aerosol samples with quantified $[INP_{-15}]$, about half (56) were collected from air masses […]

P3: General suggestion – the reviewer suggests the authors to discuss how their INP-15 generally compares to other, previous precipitation INP studies (e.g., Petters and Wright, 2015; https://doi.org/10.1002/2015GL065733) before jumping onto nX vs. INP-15. The reviewer understands that the authors intended to be straight on the point (and appreciate the concise, right on the point manuscript length to some extent), but the readers would appreciate this extra information to generalize/digest information at their end, in the reviewer's opinion.

Our concentrations were lying in the lower half of the spectrum of precipitation samples summarized by Petters and Wright (2005).

(P2L49; **P4L102**) We found cumulative concentrations of atmospheric INPs active at −15 °C ([INP$_{-15}$]) that are lying within the lower half of values summarised in Petters and Wright (2015).

P3: How SD-rich IN efficiency compares to Ullrich et al., 2017 or Niemand et al., 2012? The authors can estimate n_s and do comparisons?

We think this comparison would go beyond the scope of this study as Ullrich et al., 2017 or Niemand et al., 2012 are purely laboratory-based parametrisations.

P3L68-69: The reviewer is lost on the "It also reveals…" part. Please clarify what it means in an intuitive manner.

To clarify, we added this sentence:

(P3L68; **P5L133**) In precipitating air masses, the ratio between [INP$_{-15}$] and [n$_{2.0}$] is usually larger than in non-precipitating air masses. This reveals that the aerosol population is enriched with INPs active at moderate supercooling during precipitation, consistent with previous findings (Bigg and Miles,1964; Huffman et al., 2013).

P3L71-75: Speculative sentences – Many questions came to the reviewer's mind -What was the influence of local thermodynamics & meteorology (esp. wind spd.)? Was a proper inlet used to eliminate the impact of local turbulence etc.? Chance of resuspended particles getting into an impinger while high-volume sampling? Any hind sight 20/20 situations?

We are not quite sure whether we understand this comment correctly. We think the issue mentioned were raised already before and we hope to have answered them in different replies above.

P4L87-88: Cannot disagree – a wider spatiotempral coverage is indeed needed.

O.k. We elaborated this conclusion in the Result and Discussion section:

(P3L82; **P5L147**) To further develop a parametrisation valid for temperatures >−20 °C, we suggest to further investigate the presented functions, because INPs active at other temperatures or at other locations and during different seasons may also be associated with other particle sizes or other INP concentrations. Especially the addition of INPs in precipitating air masses should be constrained with data from all over the globe.

P7: Concentrations of ice nucleating particles active at -15 C or warmer [INP-15] –> Cumulative INP concentrations estimated at -15 C, [INP-15]

O.k. We used "cumulative concentrations of ice …" as we wanted to explain again the abbreviation for INP.

Fig. 1: Show correlation coefficients for fits. Add fits & Rs in Fig.1g, too. Discuss these in P7.

We added the functions into Fig. 1g (Fig. 3) and the correlation coefficients for the fits in a table (Table 1). We discussed the correlation coefficients. Furthermore, we added comparisons of predicted versus measured [INP$_{-15}$] for different prediction options in the supplement (Fig. S3).

(P3L82; **P5L145**) Overall, for each air mass class, the correlation coefficient of the obtained functions is equal or higher with [$n_{2.0}$] as a predictor than with [$n_{2.0}$] (Table 1). This confirms that [$n_{2.0}$] is a more powerful predictor of INPs$_{-15}$ than [$n_{0.5}$] when combined with air mass differentiation (Fig. S3).

(P4L85; **P6L156**) However, choosing the actual size range of INPs$_{-15}$ for the parametrisation can further improve the predictions. An even greater improvement in predictions is possible when we additionally distinguish between air masses that are precipitating, non-precipitating and carrying a substantial fraction of Saharan dust. More of the variance can be explained by aerosol concentrations in air masses that were non-precipitating or carrying desert dust as compared to air masses that were precipitating.

(Fig. 1g-h; **Fig. 3**)

[revised manuscript text omitted]

---

## Referee Report (RR1)

Review 2 of "Towards parametrising atmospheric concentrations of ice
nucleating particles active at moderate supercooling" by C. Mignani et al.

**General Comments**

I greatly appreciated the detailed responses to my comments, and the subsequent revisions. I
have a few issues of concern still, primarily in further discussing the superposition of the D15
parameterization on plots using aerodynamic diameter as the basis, and the *cf* factor applied in
using that parameterization. In combination, these two factors will, I believe, bring the
parameterization nearly fully in line with the observations under SD conditions. This is as it
should be, if the parameterization has any validity. One could imagine that it should fail in
capturing all dust INPs because it does not fully capture those active larger than 2.5 microns
(aerodynamic), but I have two comments about this conjecture. First, the D15 relationship was
developed from both free tropospheric (elevated from the ground) and laboratory data for which
there was a strong relation with >500 nm particle concentrations under situations that were
totally composed of, or strongly perturbed by, desert dust. In fact, I think that the present study
finds the same thing! Note the equivalent correlation coefficient for your linear regressions in SD
situations, for INPs versus 0.5 and 2.0 micron aerosol concentrations. This would not be the case
if INP concentrations under SD situations were dominated by particles larger than 2.5 microns,
would it? This brings me to the second point. The present studies say nothing about the actual
size of INPs at the site used in this study. Some investigators have found larger INPs at surface
sites, albeit likely because of influences of INPs that are not mineral dust, and partly because the
measurements are focused in the near-surface boundary layer where larger particles are always
found. Mineral dust is clearly not the major contributor at the site in this paper either, except
under SD scenarios. What the current study does show is that in order to obtain a clean relation
between aerosol concentrations and INP concentrations under most situations, one must
reference a larger particle size for parameterization. That does not permit an assured conclusion
that the INPs are always typically larger than 2 microns. You said it yourself that INPs are but a
minute fraction of the total aerosol. It could simply mean that other factors enter to populate the
smaller size ranges with particles that are not INPs and do not vary in-kind with them. This is
another important distinction in my opinion. I feel that you confuse the issue in deciding that
larger INPs are the reason that you have to go out to 2 microns to get a good correlation.

**Specific Comments**

1) Regarding changes made about previous literature on the role of ice formed and growing at
temperatures above about -15°C, I have a suggestion. The meaning of the revised statement that
"…although other temperatures would benefit from future investigations" is somewhat
ambiguous. I think you could say that investigations would benefit from relation of
measurements to overall cloud thermal structure, which may at times include lower cloud top
temperatures.

2) Related to making clear that D15 is strictly for application on mineral dust dominated
populations, somewhere around the introduction of Equation 1 it needs to be stated that the
equation will here be applied to all particles at sizes larger than 500 nm. I understand that the
concentration parameter is explicitly defined in Equation 1, but I mean that it should be said in

words that although the parameterization is strictly for mineral dusts, it will be applied to all particles. The reason for doing this is to note later that one only expects this parameterization to be valid as related to data under strong dust influences (e.g., SD here).

3) The justification for $cf = 1$ is incorrect, I believe. You say that no calibration factor is required "because INPs were observed in immersion mode (via a drop freezing assay) and not for instance, in a continuous flow diffusion chamber, where…only part of the INPs passing the instrument may become immersed in liquid droplets." This is exactly why $cf = 3$ is needed. If one were comparing data directly from a CFDC to the parameterization, then $cf = 1$ is what you would want to use, as was done in that paper in 2015. If one is using the parameterization in a model and applying it to the dust distribution, or if one is comparing to a method that captures all dust INPs active by immersion freezing, then $cf = 3$ is what one wants/needs to use. It is the full intention of the parameterization, based strongly in the results presented in that paper.

4) Regarding the explanation of the upper bound of concentrations, was dilution not possible, or simply determined not to be desirable? Perhaps say that dilution was not used in order to extend the upper range.

5) Regarding the use of aerodynamic diameter because there is no way to know shape factors and density, I want to stress again that for the purpose of showing the D15 prediction, this paper should be focused on comparison to mineral dust dominated cases. Hence, while I agree with the fact that this would be difficult for application to all unknown particle types, I think there is quite a bit known and often assumed for dust relevant shape factors and densities. So the statement that "If actual particle densities were mostly > 1 g cm$^{-2}$, our [$n_{0.5}$] would be somewhat higher than if they would have been calculated as physical particle diameters" is unsatisfying. Without showing the size distribution, one does not know how particle numbers fall off with size. Could it amount to a factor of 2? For SD, I think it easily could. For SD especially, I think that shape and density factors are probably fairly well constrained, and a 542 nm aerodynamic diameter could easily be 380 nm. It is simply a misapplication of the parameterization to use aerodynamic diameter in it and compare to aerodynamic number concentrations. If you want to argue that aerodynamic diameter is more suitable for parameterizations, that is another matter. Using a relevant assumed shape factor (perhaps 1.3) and density (perhaps 2.6) for SD in order to correct the parameterization to aerodynamic size and concentration space, this will move the D15 line to the right in the plots. How much? Using your data for SD number concentrations at two aerodynamic sizes and assuming that a linear slope exists between them (possibly not a good assumption, especially for estimated smaller size particle concentrations where the distribution may be steeper), I estimate a 2 factor concentration push of the D15 curve to the right (i.e., 2 per liter where you have plotted 1 per liter).

6) When you consider the above conservative estimate of how the D15 curve needs to be pushed to the right, and the fact that you should be using cf = 3, I would judge that your data for SD episodes are completely in line with D15 in Figure 3. I am less sure how to fix Figure S3, but perhaps that is fixed if the D15 number predictions are fixed. This means as well that the D15 parameterization  grossly under-predicts INPs in other standard conditions. This makes total sense to me for situations where biological or other INPs dominate.

Making these changes is simple, in my opinion, and then the D15 comparison only focuses on dust scenarios, and the other significant results in the paper (larger size relation to total INPs needed for this and possibly other sites, and the apparent role of biological INPs) remain unimpeded by this focus on a parameterization that does not account for them.

---

## Author Response (AR2)

We would like to thank Paul DeMott and the anonymous Reviewer for having read the revised paper. We sincerely appreciate the insightful second review by Paul DeMott. We found it very helpful to further improve the revised version of the manuscript. Point-by-point replies to the comments are below.

For clarity and easy visualization, the Referee's comments are shown from here on in black.

The authors' replies are in blue font with an increased indent below each of the referee's statements.

The relevant changes in the revised manuscript are below in green. If just a part was added to an existing sentence, then the added part is underlined. All line and page numbers in normal font refer to the first version of the revised manuscript. All line and page numbers **in bold** refer to the second version of the revised manuscript.

**Replies to the second review by Paul DeMott**

**General Comments**

I greatly appreciated the detailed responses to my comments, and the subsequent revisions. I have a few issues of concern still, primarily in further discussing the superposition of the D15 parameterization on plots using aerodynamic diameter as the basis, and the *cf* factor applied in using that parameterization. In combination, these two factors will, I believe, bring the parameterization nearly fully in line with the observations under SD conditions. This is as it should be, if the parameterization has any validity. One could imagine that it should fail in capturing all dust INPs because it does not fully capture those active larger than 2.5 microns(aerodynamic), but I have two comments about this conjecture. First, the D15 relationship was developed from both free tropospheric (elevated from the ground) and laboratory data for which there was a strong relation with >500nm particle concentrations under situations that were totally composed of, or strongly perturbed by, desert dust. In fact, I think that the present study finds the same thing! Note the equivalent correlation coefficient for your linear regressions in SD situations, for INPs versus 0.5 and 2.0 micron aerosol concentrations. This would not be the case if INP concentrations under SD situations were dominated by particles larger than 2.5 microns, would it? This brings me to the second point. The present studies say nothing about the actual size of INPs at the site used in this study. Some investigators have found larger INPs at surface sites, albeit likely because of influences of INPs that are not mineral dust, and partly because the measurements are focused in the near-surface boundary layer where larger particles are always found. Mineral dust is clearly not the major contributor at the site in this paper either, except under SD scenarios. What the current study does show is that in order to obtain a clean relation between aerosol concentrations and INP concentrations under most situations, one must reference a larger particle size for parameterization. That does not permit an assured conclusion that the INPs are always typically larger than 2 microns. You said it yourself that INPs are but a minute fraction of the total aerosol. It could simply mean that other factors enter to populate the smaller size ranges with particles that are not INPs and do not vary in-kind with them. This is another important distinction in my opinion. I feel that you confuse the

issue in deciding that larger INPs are the reason that you have to go out to 2 microns to get a good correlation.

> We address all the general concerns raised above in their more specific versions below. Only the "conclusion that the INPs are always typically larger than 2 microns" is not taken up again as a specific comment. Therefore, we address it here.

> We reformulated what we think is the contested sentence (i.e. "However, choosing the actual size range of INPs$_{-15}$ for the parametrisation can further improve the predictions.") in the Conclusion section as follows:

>> (P6L156; **P6L163**) However, relating [INPs$_{-15}$] to the number concentration of larger particles can further improve the predictions, which is not to say that INPs$_{-15}$ are always in such a size range.

**Specific Comments**

1) Regarding changes made about previous literature on the role of ice formed and growing at temperatures above about -15 °C, I have a suggestion. The meaning of the revised statement that "...although other temperatures would benefit from future investigations" is somewhat ambiguous. I think you could say that investigations would benefit from relation of measurements to overall cloud thermal structure, which may at times include lower cloud top temperatures.

> We agree that this statement is somewhat ambiguous, and we like your suggestion. We rephrased this part of the sentence as you suggested:

>> (P2L32; **P2L32**) We, therefore, in this work, focus on INPs active at that temperature, although future studies would benefit from relating measurements to overall cloud thermal structures, which may at times include lower cloud top temperatures.

2) Related to making clear that D15 is strictly for application on mineral dust dominated populations, somewhere around the introduction of Equation 1 it needs to be stated that the equation will here be applied to all particles at sizes larger than 500 nm. I understand that the concentration parameter is explicitly defined in Equation 1, but I mean that it should be said in words that although the parameterization is strictly for mineral dusts, it will be applied to all particles. The reason for doing this is to note later that one only expects this parameterization to be valid as related to data under strong dust influences (e.g., SD here).

> We specified in the text that Equation 1 will be applied to mineral dust influenced particles larger than 500 nm only. Furthermore, we now focus on comparing D15 with our SD data and make only a very cautious comparison to data not strongly influenced by mineral dust (see reply below the next comment).

>> (P3L82, **P3L83**) [INPs$_{-15}$] estimates based on D15 were calculated as:
>> $$INP_T = cf * n_{0.5}{}^{\beta} * e^{\gamma*(-T)+\delta},\qquad(1)$$
>> where $\beta = 1.25$ , $\gamma = 0.46$, $\delta = -11.6$, $T$ is the temperature in degree Celsius, $INP_T$ the ice nucleation particle concentration (std L$^{-1}$) at $T$, and $n_{0.5}$ the

number concentration of aerosol predominantly consisting of mineral dust particles with a physical diameter > 0.5 μm (std cm$^{-3}$).

3) The justification for *cf* = 1 is incorrect, I believe. You say that no calibration factor is required "because INPs were observed in immersion mode (via a drop freezing assay) and not for instance, in a continuous flow diffusion chamber, where...only part of the INPs passing the instrument may become immersed in liquid droplets." This is exactly why *cf* = 3 is needed. If one were comparing data directly from a CFDC to the parameterization, then *cf* = 1 is what you would want to use, as was done in that paper in 2015. If one is using the parameterization in a model and applying it to the dust distribution, or if one is comparing to a method that captures all dust INPs active by immersion freezing, then *cf* = 3 is what one wants/needs to use. It is the full intention of the parameterization, based strongly in the results presented in that paper.

Thanks a lot for pointing that out. We indeed misunderstood the calibration factor choice in the last version of the manuscript. We rephrased this in the text and now show in Fig. 3a predictions based on three *cf* values: 3 and 1. In addition, we added a prediction based on a *cf* = 0.086, a value reported by Schrod et al. (2017) done in the Mediterranean region. We changed Fig. 3a accordingly. Furthermore, we corrected the unit of the label on the x-axis, which was erroneously the same as on the y-axis (std L$^{-1}$) to (std cm$^{-3}$).

(P3L85-L88; **P4L96**) The calibration factor *cf* accounts for so-called instrument-specific calibration and is suggested to be three (*cf* = 3) to predict maximum immersion mode atmospheric [INP] (DeMott et al., 2015). Schrod et al. (2017), who collected samples with an unmanned aircraft system in the Mediterranean region with substantial Saharan Desert dust influence, used it as a mathematical degree of freedom when fitting Eq. 1 to their observations.

(P4L120-P4L125; **P5L128**) In general, [INPs$_{-15}$] in non-precipitating and precipitating (not dominated mineral dust) air masses were higher than in mineral dust dominated air masses for the same [n$_{0.5}$] (Fig. 3a). The observed slope for SD air masses was the same as that predicted by the D15 parametrisation. The offset of the D15 curve depends on the calibration factor (*cf*, Eq. 1). Observed SD data were between the D15 curves with *cf* set to 1 and to 0.086, respectively. The latter value is reported in Schrod et al. (2017), who sampled the Saharan Dust Layer above Cyprus with a drone up to 2850 m a.s.l.

[Figure]

(Figure 3, **Figure 3**) […] The gray lines show the D15 parametrisation extrapolated to -15 °C and corrected for the difference between physical and aerodynamic diameters (see Methods section) with three different calibration factors (Eq. 1): $cf$ = 3 (dashed), cf = 1 (continuous), and $cf$ = 0.086 (dotted). The latter value was the best fit found by Schrod et al. (2017) for observations of Saharan dust above Cyprus.

4) Regarding the explanation of the upper bound of concentrations, was dilution not possible, or simply determined not to be desirable? Perhaps say that dilution was not used in order to extend the upper range.

Dilutions were not considered to be reliable because of unavoidable storage and its effects. The sampling campaign was at times extremely busy and our target was to analyse the (undiluted) samples as quickly as possible after sampling. Therefore, dilutions had to wait.

5) Regarding the use of aerodynamic diameter because there is no way to know shape factors and density, I want to stress again that for the purpose of showing the D15 prediction, this paper should be focused on comparison to mineral dust dominated cases. Hence, while I agree with the fact that this would be difficult for application to all unknown particle types, I think there is quite a bit known and often assumed for dust relevant shape factors and densities. So the statement that "If actual particle densities were mostly > 1 g $cm^{-2}$, our [$n_{0.5}$] would be somewhat higher than if they would have been calculated as physical particle diameters" is unsatisfying. Without showing the size distribution, one does not know how particle numbers fall off with size. Could it amount to a factor of 2? For SD, I think it easily could. For SD especially, I think that shape and density factors are probably fairly well constrained, and a 542 nm aerodynamic diameter could easily be 380 nm. It is simply a misapplication of the parameterization to use aerodynamic diameter in it and compare to aerodynamic number concentrations. If you want to argue that aerodynamic diameter is more suitable for parameterizations, that is another matter. Using a relevant assumed shape factor (perhaps1.3) and density (perhaps 2.6) for SD in order to correct the parameterization to aerodynamic size and concentration space, this will move the D15 line to the right in the plots. How much? Using your data for SD number concentrations at two aerodynamic sizes and assuming that a linear slope exists between them (possibly not a good assumption, especially for estimated smaller size particle concentrations where the distribution may be steeper), I estimate a 2 factor concentration push of the D15 curve to the right (i.e., 2 per liter where you have plotted 1 per liter).

Thank you for the clarification and suggestion of a shape factor and a density. Using these values in combination with the size distributions measured during SD events pushes the D15 curve downward by a factor of around 2 on Fig. 3a with aerodynamical diameter as unit on the x-axis.

(P3L88-P4L92, **P3L87**) A physical diameter of 0.5 µm is equivalent to an aerodynamic diameter of 0.9 µm, assuming a particle density of 2.6 g $cm^{-2}$ and a shape factor of 1.3 (Raabe, 1976), which are typical values for mineral dust particles. Similar transformations for observations not dominated by mineral dust would require information about densities and shapes of the main components of sampled particle populations, which were not available for our site and would require unsupported assumptions. Therefore, we chose

to show for all our observations the directly measured particle concentrations in terms of *aerodynamic diameter*. To use D15 parametrisation in our context, we corrected predicted [INP] for the difference between the aerodynamic diameter measured and the physical diameter used in Eq. 1 by multiplying $n_{0.5}$ in Eq. 1 by the ratio of particles with aerodynamic diameters > 0.9 µm (equivalent to 0.5 µm physical diameter) to particles with aerodynamic diameters > 0.5 µm, which we observed in Saharan dust dominated air masses during our campaign. The average value of this ratio was 0.59.

6) When you consider the above conservative estimate of how the D15 curve needs to be pushed to the right, and the fact that you should be using cf = 3, I would judge that your data for SD episodes are completely in line with D15 in Figure 3. I am less sure how to fix Figure S3, but perhaps that is fixed if the D15 number predictions are fixed. This means as well that the D15 parameterization grossly under-predicts INPs in other standard conditions. This makes total sense to me for situations where biological or other INPs dominate. Making these changes is simple, in my opinion, and then the D15 comparison only focuses on dust scenarios, and the other significant results in the paper (larger size relation to total INPs needed for this and possibly other sites, and the apparent role of biological INPs) remain unimpeded by this focus on a parameterization that does not account for them.

Using *cf* = 3, instead of *cf* = 1, pushes the D15 curve upward by a factor of three, while scaling D15 to the aerodynamic diameter pushes it downward by a factor of two (Fig. R1). Combined, the difference to the previous position of the curve is small (a factor of 1.5 upward).

[Figure]

**Figure R1:** Same as Fig. 3a with the D15 parametrisation not scaled to aerodynamic diameter using two different calibration factors (cf = 1, dashed line; cf = 3, dotted line) and scaled to aerodynamic diameter using cf = 3 (continuous line).

Our comparison with D15 is now focused on SD air masses using different calibration factors. Fig. S3a of the first version of the revised manuscript was dropped because it unjustifiably applied D15 to air masses for which it was not developed. Fig. S3 is adapted accordingly.

[Figure]

(Figure S3, **Figure S3**). Measured and predicted cumulative concentrations of ice nucleating particles active at -15°C [$INP_{-15}$] (std $L^{-1}$) for (a) prediction based on a single trendline fitted through all data of aerosol particles with aerodynamic diameters > 0.5 µm [$n_{0.5}$], (b) predictions based on [$n_{0.5}$] and three different trendlines fitted through the data of PRECIP (blue circles), NON-PRECIP (green triangles), and SD (red squares) air masses, and (c) same as (b), but based on aerosol particles with aerodynamic diameters >2.0 µm [$n_{2.0}$]. Shapes in (a) are consistent with those in (b). However, they are coloured in gray as the prediction is independent of air mass classes. A range of a factor of two (dotted lines) about the 1:1 line (solid line) as well as the percentage of values lying within that range are shown in all panels.

Last, we added a few words to a sentence in the Conclusion, which was erroneously not that precise.

(P6L158; **P6L167**) The absolute value of additional INPs in precipitating air masses, versus non-precipitating air masses, seems to be independent of total aerosol concentrations.

**References**

Raabe, O. G.: Aerosol aerodynamic size conventions for inertia! Sampler calibration, Journal of the Air Pollution Control Association, 26, 856–860, 1976.

Schrod, J., Weber, D., Drücke, J., Keleshis, C., Pikridas, M., Ebert, M., Cvetkovi´c, B., Nickovic, S., Marinou, E., Baars, H., Ansmann, A., Vrekoussis, M., Mihalopoulos, N., Sciare, J., Curtius, J., and Bingemer, H. G.: Ice nucleating particles over the Eastern Mediterranean measured by unmanned aircraft systems, 17, 4817–4835, 2017.

---

## Author Response (AR3)

We have updated the data availability section with the final link towards the website where the data are published and added the DOIs for each reference.